# Accelerating flash droughts induced by the joint influence of soil moisture depletion and atmospheric aridity

Yamin Qing[1,5], Shuo Wang [1,2,5 ✉], Brian C. Ancell[3] & Zong-Liang Yang[4]

The emergence of flash drought has attracted widespread attention due to its rapid onset. However, little is known about the recent evolution of flash droughts in terms of the speed of onset and the causes of such a rapid onset phase of flash droughts. Here, we present a comprehensive assessment of the onset development of flash droughts and the underlying mechanisms on a global scale. We find that 33.64−46.18% of flash droughts with 5-day onset of drying, and there is a significant increasing trend in the proportion of flash droughts with the 1-pentad onset time globally during the period 2000−2020. Flash droughts do not appear to be occurring more frequently in most global regions, just coming on faster. In addition, atmospheric aridity is likely to create a flash drought-prone environment, and the joint influence of soil moisture depletion and atmospheric aridity further accelerates the rapid onset of flash droughts.

[1] Department of Land Surveying and Geo-Informatics and Research Institute for Land and Space, The Hong Kong Polytechnic University, Hong Kong, China. [2] Shenzhen Research Institute, The Hong Kong Polytechnic University, Shenzhen, China. [3] Department of Geosciences, Texas Tech University, Lubbock, TX, USA. [4] Department of Geological Sciences, The John A. and Katherine G. Jackson School of Geosciences, University of Texas at Austin, Austin, TX, USA. [5] These authors contributed equally: Yamin Qing, Shuo Wang. ✉email: shuo.s.wang@polyu.edu.hk

Flash droughts have been receiving increasing attention from the scientific community in recent years. Compared with traditional, more slowly developing droughts, flash droughts evolve with a rapid onset development along with a fast depletion of water availability that may cause an imbalance of ecosystems and agricultural systems[1,2]. Since flash droughts can be distinguished by their rapid development, there is less early warning for impact preparation, potentially causing more severe impacts on agriculture and society than the slowly evolving droughts[3,4]. For example, the 2012 flash drought that occurred over the Central United States caused a tremendous impact on agricultural production and the economy, with about $35.7 billion losses attributed to this event[5]. Furthermore, analysis of retrospective climate simulations found virtually no dry signal, owing to the unusual speed and intensity (developed in May and reached peak intensity by August), over the major corn-producing regions of the eastern Great Plains where this drought event resulted in major curtailment of corn crop yields. The flash drought event that occurred over Southern Queensland, Australia in 2018 de-vegetated the landscape and drove livestock numbers to the lowest level in the country[6]. And some countries experienced more frequent flash droughts in recent years, including southern China[7–9], USA[10,11], and Africa[12].

There is currently a lack of a consistent definition of flash droughts. The definitions proposed in previous studies can be generally classified into two types. One is based on the duration of flash droughts. Flash droughts as proposed by Mo and Lettenmaier[13,14] used different combinations of thresholds of soil moisture (SM), temperature (T), precipitation (P), and evapotranspiration to identify the heat wave flash drought (HWFD) and the P deficit flash drought (PDFD). Another definition is based on the rapid intensification rate of flash droughts[4,9,15,16]. These two types of definitions of flash droughts provide us with alternative approaches to assess this extreme phenomenon from different perspectives (duration and intensification rate). In addition, there are a variety of indices available to identify flash droughts, including the Evaporative Demand Drought Index (EDDI)[17,18], the Evaporative Stress Index (ESI)[19,20], the Standard Evaporative Stress Ratio (SESR)[21], the combination of the ESI and the Rapid Change Rate Index (RCI)[22–24], P[25], vegetation[26], and SM[9,27–29]. Several definitions were also proposed based on the classification (percentile/quantile) system of US Drought Monitor (USDM)[30–32]. Previous studies have indicated that SM anomalies are useful for characterizing the drought onset, particularly for rapid-onset droughts[33,34]. The rapid decline in SM could potentially serve as a precursor for flash droughts[4]. Osman et al.[35] examined the key climate variables used in flash drought definitions, including P, root-zone SM, T, as well as actual and potential evapotranspiration. And they find that the root-zone SM shows the clearest signal when flash droughts occur.

The occurrence of flash droughts has been examined in many regions worldwide, but most studies are limited to a basin[28,36,37], several states[21,30,38], or a country[9,13,29,39]. A previous study produced a global map of flash droughts, with a focus on the contribution of evapotranspiration (ET) and P to the occurrence of flash droughts[40]. Nonetheless, few studies explore the rapid onset phase of flash droughts and underlying causes of the rapid onset which is the most important characteristic of flash droughts. In addition, local and regional extreme events are conditioned by large-scale atmospheric circulation that can affect the onset of flash droughts under potential processes including upper-level ridges, land–atmospheric interactions, and monsoons[32,41,42]. These processes influence the climate characteristics and the occurrence of flash droughts over different regions around the world. Thus, a global picture of flash drought onset is needed to reveal the spatial pattern and temporal variability of the speed of flash drought development, advancing our understanding of flash droughts on a global scale.

Flash droughts can be viewed as a subset of droughts that are distinguished from slowly developing droughts by their rapid onset development[4]. The rapid onset development must include not only a rapid onset but also a rapid rate of intensification. The rapid onset is a turning point, which depicts a shift from non-drought to drought conditions, but it lacks an interpretation of the process of change. The rapid intensification rate can characterize either the onset or the later phase of drought events, and thus it can depict a developed drought becoming more severe but cannot represent the beginning of a flash drought event. Even though it is recognized that flash droughts evolve rapidly during the onset development, it remains unclear how fast flash droughts evolve. Does the rapid development take place within one week, one month, or longer time? On the other hand, few studies address the causes of the rapid onset speed which is the most important characteristic of flash droughts. Thus, exploring the onset development timescales and identifying the drivers that may trigger and speed up the onset of flash droughts can provide insights into the prediction of flash droughts and the development of early warning systems for mitigating the impacts of flash droughts.

Here, we explore the rapid onset development of flash droughts identified by taking into account both flash (a rapid onset accompanied by a rapid intensification rate) and drought severity (SM drops below a specific threshold for a period of time) on a global scale. Specifically, we examine the spatial pattern of the timescale of flash drought onset phase and reveal how fast flash droughts develop over the past 21 years from 2000 to 2020. This global study is to shed light on the onset development timescales and the causes of the rapid onset speed, advancing our understanding of the evolution tendency of flash droughts and providing insights into the implementation of flash drought forecasts and early warning systems.

## Results

**Identification of global hot spots of flash droughts**. We compared the mean variations in SM percentiles for flash drought events of all grid points, identified by the intensification rate (Fig. 1a) and the duration (Fig. 1b) of flash droughts. To satisfy the drought condition, SM should decrease to a critical level (the 20th percentile threshold, which is recommended by the U.S. Drought Monitor) because SM below such a threshold indicates a start of abnormally dry conditions. For the definition from a perspective of the intensification rate, the mean SM from all three datasets should reach below the 20th percentile throughout the same days over the period of 2000 to 2020. By contrast, based on the duration-based definition, about 17.78−23.77% and 6.75−8.28% of the events detected would be viewed as non-drought or non-moderate events for HWFD and PDFD since SM only decreases from above the 40th percentile to above the 20th percentile. Particularly for HWFD, the mean variations of SM percentiles of all flash droughts from all three datasets are between 20th and 30th percentiles (Fig. 1b). As for another feature (flash), the events identified by the intensification rate-based definition present a sharp decrease in SM, on average, from the 52.76−64.12th percentile to the 25.40−39.37th percentile (from the highest point to the lowest point), with a decline rate of larger than the 5th percentile per pentad. In comparison, only 0.20−1.15% and 0.10−1.06% of the events identified by the duration-based definition experience a rapid decline (SM decreases from the 40th percentile to below the 20th percentile) in SM for HWFD and PDFD over the period of onset development phase. And they decrease, on average, from the 23.91−27.27th

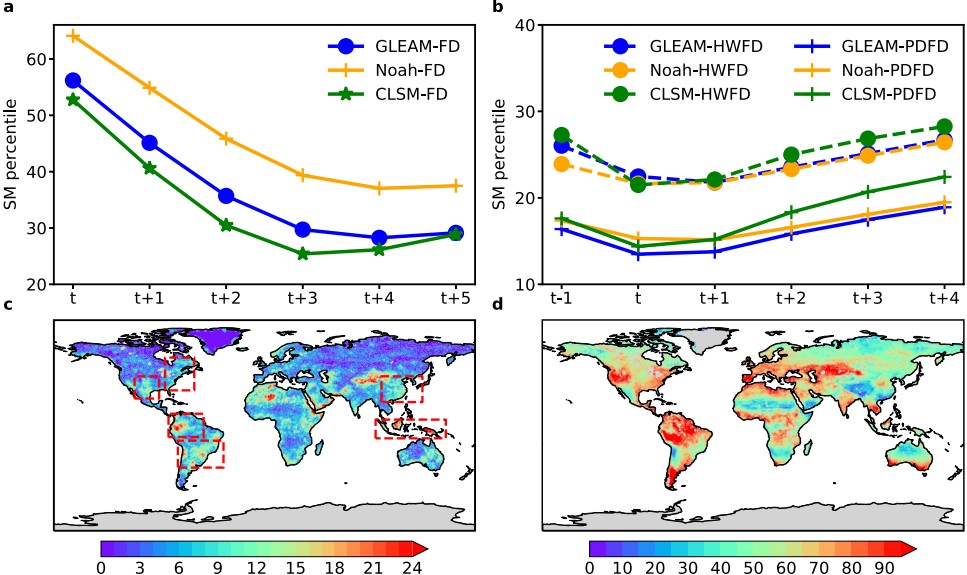

**Fig. 1 Comparison of variations in SM percentiles and mean frequencies of flash droughts according to different definitions from the perspective of intensification rate and duration. a**, **b** Temporal variations in SM percentiles for flash drought events of all grid points based on two definitions. Please note that the SM percentile represents the magnitude of the decline in SM, as shown in **a**, which was calculated for the whole time series. **c**, **d** Frequencies of occurrence of flash droughts based on two definitions. Intensification rate-based definition: SM decreases from above the 40th percentile to below the 20th percentile with an average decline rate of no less than the 5th percentile for each pentad, and the SM below the 20th percentile should last for no <3 pentads. If the declining SM rises up to the 20th percentile, flash droughts will terminate. Duration-based definition: The pentad-mean surface T anomaly > one standard deviation, ET anomaly > 0, P anomaly < 0, and SM percentile < 40% for HWFD events. The pentad-mean surface T anomaly > one standard deviation, ET anomaly < 0, SM percentile < 40%, and P percentile < 40% for PDFD events. **d** Total numbers of HWFD and PDFD.

percentile to the 21.49−22.47th percentile (the lowest point) for HWFD, and from the 17.62−16.39th percentile to the 13.49−14.38th percentile for PDFD (color lines in Fig. 1b). Furthermore, some events with SM decreasing to below the 40th percentile occur, but then rapidly recover up to the 40th percentile within only one or two pentads under abnormally dry conditions (red box in Supplementary Fig. 3). These events cannot be viewed as flash droughts because only one or two pentads are inadequate to diminish crop productivity and yield. Perhaps even if the drought lasts for a short period of time (1 or 2 pentads), it may also affect the growth of crops if accompanied by high temperature and wind, but this situation is not within the scope of our discussion. Therefore, the use of the definition focusing solely on the duration of flash droughts cannot guarantee that all events satisfy the key characteristics (drought severity and flash) of flash droughts.

The frequencies of flash droughts identified by the intensification rate-based definition are lower than those detected by the duration-based definition (Fig. 1c, d). This is because the use of the duration-based definition captures more events that last for only one pentad, whereas the mean duration of events identified by the intensification rate-based definition is longer than one month (Supplementary Fig. 4). Nonetheless, both methods (definitions) indicate that flash droughts are most likely to occur in humid and semi-humid regions (e.g., Southeast Asia, East Asia, Amazon Basin, Eastern North America, and Southern South American) according to the global map of the arid, transitional, and humid regimes (Supplementary Fig. 5). This is consistent with previous studies[7,9,43]. Particularly, the hot spots for the high frequency of flash droughts are highlighted, and the contributions to the high frequency of flash droughts are related to different phenomena of atmospheric circulation including monsoons, the upper-level ridge, and land–atmosphere coupling (red boxes in Fig. 1c). The atmospheric circulation may cause the anomalies of P and T, which are likely to trigger flash drought events[44–50]. For

example, East Asia and Southeast Asia are tropical monsoon climate regions[51,52], and during the break of monsoons, the prolonged significant negative anomaly of P may lead to a rise in T, which in turn causes a rapid depletion in SM. Furthermore, the combination of significant negative P and positive T anomalies could lead to an increased atmospheric water demand, accelerating the decline in SM[29,53–55]. It should also be noted that Tibet is a drought-prone region, with more than one drought events occurring each year. With high elevations on the Tibetan Plateau, terrestrial ecosystems in this region are vulnerable to climate change. Although this region is also affected by monsoons from East Asia and Southeast, it is obviously dominated by the plateau alpine climate[56,57].

**Assessment of onset development timescales of flash droughts.** To facilitate further understanding of flash drought characteristics, we divided the whole world into 21 regions for conducting an in-depth regional analysis of flash droughts (Supplementary Table 1). To conduct a comprehensive assessment of the onset time of flash droughts, we divided flash droughts into five different types according to the longest possible onset development phase of flash droughts (≤1 month was proposed by Otkin et al. [4]), including 1 pentad, 2 pentads, 3 pentads, 4 pentads, and 5 pentads. Please note that the onset development phase refers to a rapid onset of drying before drought conditions occur. Figure 2 compares the percentages of flash droughts at different onset times relative to all flash droughts as well as their annual evolutions during the entire study period of 2000–2020. In general, 33.64−46.18% of flash droughts developed in 1 pentad, especially in South Asia, Central Asia, Western Africa, and Amazon Basin. About 20.12−28.17% and 19.32−32.53% of flash droughts occurred in 2 and 3 pentads, respectively. In comparison, there are 5.34−11.22% and 0.99−2.26% of flash droughts developing in 4 and 5 pentads, respectively. In general, more than 70% of flash droughts develop

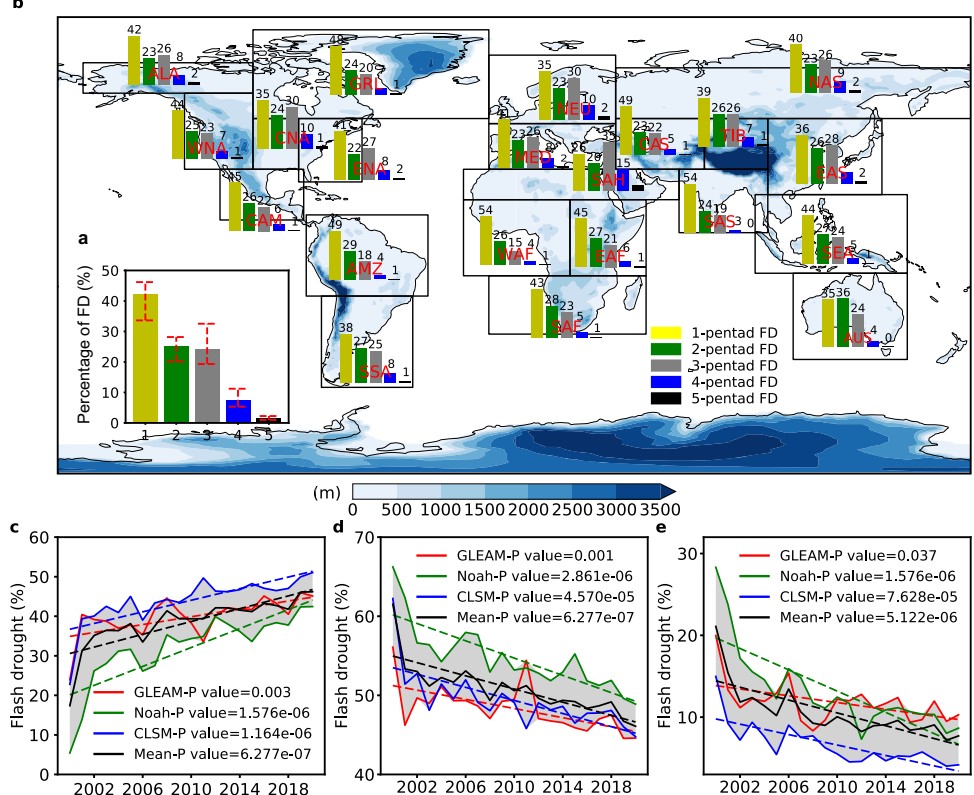

**Fig. 2 Comparison of percentages of flash droughts at different onset times relative to all flash droughts and their temporal evolutions. a** Global proportion of flash droughts at different onset times relative to all flash droughts. The red dashed lines at the top of each bar represent the range of uncertainty in three different datasets. **b** Proportion of flash droughts at different onset times relative to all flash droughts over 21 regions. **c–e** Temporal evolutions of percentages of flash droughts with the onset times of 1-, 2-3, and 4-5 pentads relative to all flash droughts. The linear annual trends in the proportion of flash droughts at different onset times are estimated based on the Sen's slope estimator, and statistical significances in trends are determined based on the MK test for the entire study period (2000−2020). The shaded areas represent the range of uncertainty in three different datasets.

within half a month (≤3 pentad), and more than 30% of flash droughts develop only within 5 days (≤1 pentad) accompanied by a high intensification rate (≥5th percentile), whereas the traditional droughts may take 5–6 months to develop due to the cumulative effects (e.g., P deficit) of related climate variables[58].

We used the nonparametric Mann–Kendall statistic to examine changes in the percentages of flash droughts at different onset times relative to all flash drought events at an annual timescale. Both the trend and the number of flash drought events developing in 2 and 3 pentads, derived from all three models, are consistent and match well with each other. In addition, the number of flash droughts developing in 5 pentads is too small to perform statistical significance tests over a 21-year period, and thus we incorporated the events at 2- and 3-pentad onset times into one category (flash droughts developing in 2-3 pentads) and the events at 4- and 5-pentad onset times into another category (flash droughts developing in 4-5 pentads). We find that the proportion of flash droughts with 1-pentad onset of drying in all flash drought events shows a significant (P < 0.01) increasing trend globally during the whole period 2000−2020 for all three datasets. The magnitude of estimated slope suggests an increase of 3.23−19.03% in the proportion of 1-pentad onset flash droughts during the study period for all three datasets, with an annual growth of 0.15−0.91% (Fig. 2c). Over most regions, the 1-pentad onset flash droughts also show an increasing trend, especially in South Asia, Southeast Asia, and Central North America where the increase in the proportion of 1-pentad onset flash droughts is greater than 20% (Supplementary Fig. 6), indicating that more flash drought events develop within one week over these regions.

By contrast, the proportions of flash droughts developing both in 2-3 and 4-5 pentads, derived from all three datasets, show a statistically significant (P < 0.05) decrease from 56.08−66.25% and 14.98−28.32% in 2000 to 44.60−48.90% and 4.17−10.29% in 2020, implying that flash droughts take less time to develop (Fig. 2d, e). Similar trends can also be found in most sub-regions, with the largest decreasing trend over Southeast Asia, followed by South Africa, South Asia, and Amazon Basin (Supplementary Figs. 7, 8). More importantly, we find that there is no consistently significant increasing trend in the frequency of all flash droughts globally based on three datasets (Supplementary Fig. 9). Over most regions the frequency of flash droughts, in general, does not show a statistically significant trend, and there is only a significant trend over the Mediterranean Basin, Central Asia, and Sahara (Supplementary Fig. 10).

**Physical processes that cause flash droughts.** It is widely accepted that the rapid onset of flash droughts is caused by moisture imbalance that is related to P and T[32]. The increase (decrease) in P can lead to the corresponding increase (decrease) in SM, and the increase (decrease) in T also contributes to the decrease (increase) in SM. Thus, P and T are the key factors influencing the occurrence of flash droughts. More importantly, the simultaneous occurrence of extreme high T and P deficit may exacerbate the depletion of SM to develop a flash drought event. For example, Queensland experienced the extreme high T coinciding with an extreme deficit of P in January 2018, which caused a flash drought event lasting from January to June in 2018[6]. In

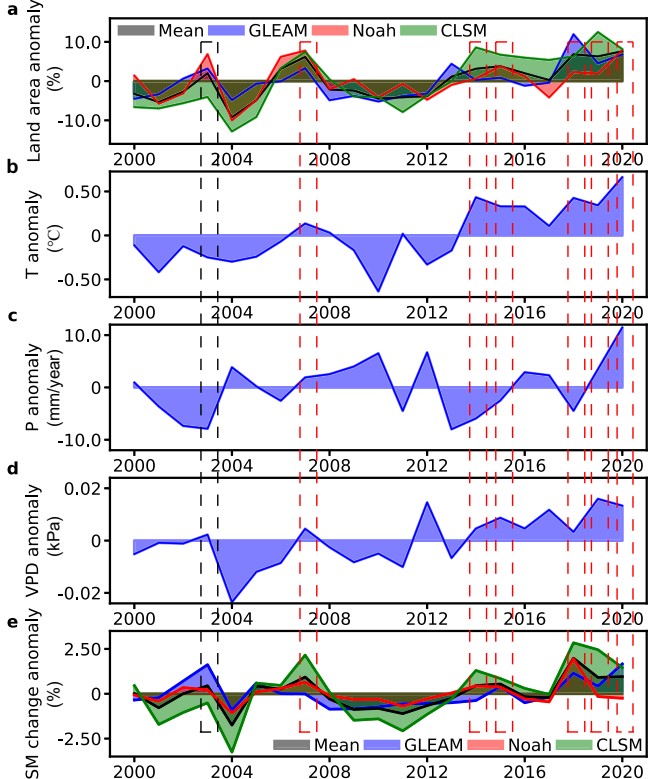

**Fig. 3 Interannual variations of flash drought-affected land area and its related variables over the NEU-MEU from 2000 to 2020. a, e** Anomalies of flash drought-affected land area and relative SM change over the NEU-MEU from three different datasets. The relative SM change is defined as the change of SM percentiles between adjacent pentads (current and next-pentad SM percentiles), where positive values represent a decline in SM at the current pentad. **b–d** Anomalies of annual mean T, annual total P, and annual mean VPD over the NEU-MEU.

addition, vapor pressure deficit (VPD) dictates the evaporative demand imposed on plant leaves[59], which can enhance the atmospheric water demand and exacerbate the SM depletion.

To explore flash drought evolution characteristics and to conduct attribution analysis, we plotted the time series of anomalies of annual flash drought-affected land areas, annual mean T, annual total P, annual mean VPD, and relative SM change over the NEU-MEU (the total area of Northern Europe and Mediterranean Basin) during 2000–2020 (Fig. 3). These two places cover most of the European continent, and the annual frequencies of flash droughts of Northern Europe and Mediterranean Basin show an increasing trend (Supplementary Fig. 10). It is found that high T, low P, and high VPD often coincide with a high percentage of flash drought-affected land areas. The percentages of flash drought-affected land areas in 2007, 2014, 2015, 2018, 2019 and 2020 rank in the top 6 during 2000–2020 (red boxes in Fig. 3), and most of them are accompanied by high VPD, warm T anomaly, and P deficit (Fig. 3a−d). On the other hand, there are few significant correlations between T and the percentage of flash drought-affected land areas. The high percentage of flash drought-affected land areas is not always accompanied by high T (black box in Fig. 3). This suggests that the flash drought occurrence is not dominated by T. The high T can cause SM to drop, but such a decrease in SM anomalies is not large[14]. The VPD and T during onset are higher and the P are lower than before and recovery phases of flash droughts, which provides the conditions more favorable for the occurrence of flash droughts (Supplementary Fig. 11). In addition, the variation of

annual mean VPD is similar to the variation of relative change in annual mean SM. Specifically, the relative change in SM increases (decreases) when VPD increases (decreases), suggesting that the increasing VPD triggers a rapid decline in SM. Furthermore, we find that the significant ($P < 0.05$) increasing trend in the percentage of 1-pentad onset flash droughts over NEU is accompanied by a significant increasing trend in annual mean VPD, whereas MED experiences an insignificant ($P > 0.05$) increase in the percentage of 1-pentad onset flash droughts accompanied by an insignificant increase in annual mean VPD (Supplementary Figs. 12, 13). Thus, high T, low P, and high VPD create a flash drought-prone environment, and VPD is a significant factor influencing the speed of onset, which tends to drive a rapid decline in SM to cause flash drought events.

There is a strong negative correlation between SM and VPD, and the low SM is always accompanied by the high VPD[60]. In turn, the high VPD enhances the atmospheric evaporative demand, which can exacerbate the decline in SM through land–atmosphere interactions[61]. To investigate the contribution of VPD to the occurrence of flash droughts, we examined the underlying relationship between SM and VPD (Fig. 4). We find that SM is strongly correlated with VPD, which is evident in the bimodal distribution of SM and VPD: high VPD and low SM, and low VPD and high SM (Fig. 4b). To further explore the onset phase of flash droughts, we defined the bivariate extremes as those with SM below the 30th percentile (the 30th percentile was used to represent the mean condition of flash drought onset phase) and VPD above the 90th percentile. Probability multiplication factor (PMF) was calculated to assess the increasing frequency of compound events compared to those expected when SM and VPD were independent, and a PMF of 1 represents no increase in the joint probability. We find that certain regions with a high value of PMF are consistent with those experiencing a high frequency of flash drought events (dashed boxes in Figs. 1c and 4a). Even if the thresholds of SM and VPD change, similar patterns can also be obtained (Supplementary Fig. 14). We also calculated the PMF of SM-T and SM-P, and the corresponding patterns show less correlation with those of flash drought frequency compared with SM-VPD (Supplementary Fig. 15). More importantly, a positive correlation ($R = 0.63$, the points of Tibet and Sahara were excluded from correlation analysis because the PMF values of almost all points are <1, suggesting that there is no increase in the co-occurrence probability) between the frequency of flash droughts occurring in 19 regions and PMF is found (Fig. 4c), especially for the Southeast Asia, Amazon Basin, and Eastern North America where the joint probability of bivariate extremes is ~1.77–2.36 times higher than the probability of the independent combination. This indicates that the strong coupling of SM and VPD through land–atmosphere feedbacks contributes to the rapid decline in SM. Moreover, the positive correlation between the frequency of flash droughts occurring in 19 regions and PMF will increase to 0.65 when the threshold of extreme SM decreases to the 20th percentile, suggesting that the intensity of SM-VPD coupling will increase when there is a further decline in SM (Fig. 4d).

T, P, and VPD are all related to the atmospheric demand for water (an indicator of atmospheric aridity) that affects photosynthesis and transpiration of plants and therefore play a critical role in regulating water of terrestrial ecosystem[62,63]. And higher atmospheric aridity (high VPD, low P, and high T) means greater potential to pump moisture out of soils and plants. The development of flash droughts can be accelerated when SM depletion co-occurs with atmospheric aridity. This is the reason why most flash drought-affected areas experience rising VPD and T as well as decreasing P. Specifically, VPD plays an important role in the land–atmosphere interaction, which has a strong

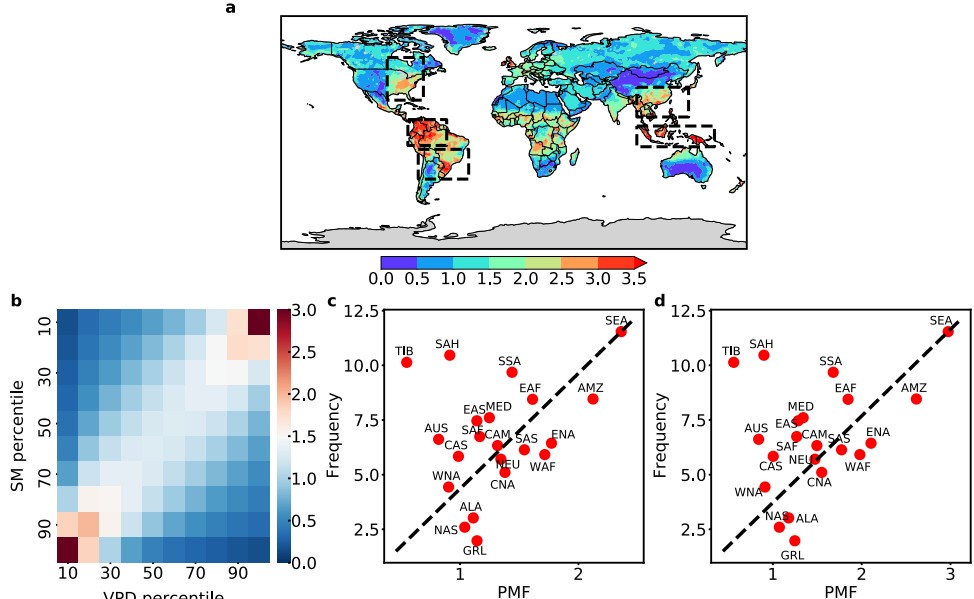

**Fig. 4 Relationship between SM and VPD. a** Mean PMF of concurrent extreme SM (below the 30th percentile) and VPD (above the 90th percentile) for the period 2000–2020 based on three datasets. **b** Mean probability (%) of percentile bins of SM and VPD across all land grid points. **c** Spatial relationship between the frequency of occurrence of flash droughts and the PMF of SM and VPD extremes (below the 30th percentile SM and above the 90th percentile VPD). **d** Spatial relationship between the frequency of occurrence of flash droughts and the PMF of SM and VPD extremes (below the 20th percentile SM and above the 90th percentile VPD). Each point in **c** and **d** is the mean of the results from three datasets.

negative correlation with SM. According to our study, the variation of VPD is almost consistent with the variation of relative change in SM, indicating that VPD dominates the change in SM, and such a change in SM can be enhanced by the increasing SM-VPD coupling through land–atmosphere feedbacks. Therefore, atmospheric aridity creates a perfect condition for the occurrence of flash droughts, and the joint influence of SM depletion and atmospheric aridity further enhances the rapid onset of flash droughts.

## Discussion

It should be noted that the frequency of occurrence of flash droughts is largely affected by the thresholds of SM chosen to identify flash droughts. We compared the frequencies of occurrence of flash droughts under different percentile thresholds of SM (Supplementary Fig. 16). We find that the spatial patterns are similar under different thresholds, but the number of detected flash droughts varies greatly. When lifting the SM percentile to the 50th percentile, the number of detected flash droughts is too small to carry out tests for statistical significance over most regions of the world for the period 2000−2020. More flash drought events can be identified when the threshold decreases to the 30th percentile, but such a decline (≤ the 5th percentile) in SM from the 30th to the 20th percentile may not be viewed as a rapid onset under the worst-case scenario (one month). Generally, when the threshold of SM is changed to 30th and 50th percentiles, there is a significantly increasing trend in the percentage of flash droughts developing in 1 pentad and a significantly decreasing trend in the percentage of flash droughts developing in 2-3 and 4-5 pentads (Supplementary Fig. 17).

We also conducted sensitivity analysis by changing soil depths. By comparing flash drought frequencies identified by SM in the top and the root-zone layers (Supplementary Fig. 18), we find that the difference in flash drought frequencies, derived based on different soil depths, is larger than the difference resulting from different SM thresholds. The use of top-layer SM is able to capture more flash drought events since the top-layer soil responds

quickly to the evapotranspiration increase and the P deficit. In comparison, the water stored in the root-zone layer is directly available to support plant growth, which is a dominant factor affecting agricultural productivity. The deficit in the root-zone SM can result in plant death and yield loss. Thus, we identified flash droughts based on the changes in the root-zone SM instead of the top-layer SM.

Flash droughts are characterized by a rapid onset, and we find that the rapid onset can be attributed to large-scale atmospheric dynamics and land–atmosphere feedbacks. The local moisture imbalance is influenced by large-scale atmospheric dynamics through changing the local T, P, and VPD. The anomalies of multiple factors (T, P, and VPD) that occur simultaneously may trigger the rapid onset of flash droughts, such as the aforementioned case of NEU-MEU. In addition, we find that the strong coupling of SM and VPD also contributes to the development of flash droughts since the dependence between extreme high VPD and low SM leads to a rapid SM depletion and thus triggers flash drought development. Furthermore, the likelihood of the accelerating onset of flash droughts increases significantly due to anthropogenic climate change. For example, Wang and Yuan[64] found that anthropogenic climate change increased the likelihood of onset speed and intensity of the 2019-like flash drought event by 24 ± 16% and 37 ± 9%, respectively. Moreover, ocean–atmosphere teleconnections may influence the development of flash droughts through atmospheric circulation associated with a rapid decline in SM in conjunction with P deficit[14]. Vegetation is also a key component affecting the rapid onset development of flash droughts because of its important role in mediating the transpiration. Crops can suffer from moisture stress more quickly by pumping water from deep soil, which may trigger a rapid development of flash droughts. Exploring mechanisms of flash droughts would be challenging but a promising direction for unraveling the mystery of flash drought onset.

**Societal implications**. Collectively, our findings suggest that flash droughts tend to occur in humid and semi-humid regions,

including Southeast Asia, East Asia, Amazon Basin, Eastern North America, and Southern South American. The identification of flash drought-prone regions and global hot spots can provide valuable insights to help inform policymakers and stakeholders on potential risks of flash droughts. Specifically, the onset development of flash droughts is becoming faster, and ~33.64−46.18% of flash droughts occur within 5 days for the period 2000−2020, which poses a great challenge for drought monitoring. There are several products and indicators used to capture flash drought events, including the U.S. Drought Monitor (USDM), the Drought Impact Reporter (DIR), the U.S. Crops and Livestock in Drought, the Climate Prediction Center (CPC) SM, and the Vegetation Drought Response Index (VegDRI). Although some of them are available on a daily or weekly basis, but not for all countries. Thus, assessing the relatively fast onset development timescales of flash droughts can provide useful information for upgrading drought monitoring systems and indicators all over the world. To further improve the ability of monitoring and predicting flash droughts, the criterion of a rapid intensification rate should be taken into account in addition to the relatively short onset timescales for capturing the unique characteristics of flash droughts (rapid onset and rapid intensification). The change in the state of the climate should also be incorporated into flash drought monitoring and prediction so that it remains meaningful in a warming climate.

It should also pay close attention to relevant compound and atmospheric circulation dynamics that may trigger the onset of flash droughts. Identifying the drivers of flash droughts and the associated factors that may speed up the rapid decline in SM is crucial to developing plausible risk mitigation scenarios based on multi-criteria analysis of potential weather conditions in different geographical contexts. According to our study, the atmospheric aridity (high T, low P, and high VPD) is conducive to flash drought development, and the resulting anomalies are likely to trigger the occurrence of flash droughts. Specifically, VPD is a significant factor causing the rapid decline in SM, and the SM–VPD coupling can further enhance the SM depletion. Thus, we should pay more attention to the regions with relatively high risk of compound extreme events, especially for compound drought and heatwave events. In addition, we should focus on the regions with strong land–atmosphere coupling since there is a high correlation between the joint probability of bivariate extremes and the frequency of flash droughts. To advance the understanding of flash droughts, more research efforts should be made, especially for quantifying environmental, social and economic vulnerability to flash droughts and their impacts.

## Methods

**Definition of flash droughts**. To robustly identify flash droughts, we took into account both the rapid intensification rate and the drought condition in this study: the pentad (5 days) indicates the root-zone SM decreasing from above the 40th percentile to below the 20th percentile, with an average decline rate of no less than the 5th percentile for each pentad. If the declining SM rises up to the 20th percentile again, flash droughts will terminate. It should be noted that the 40th and 20th percentiles were determined throughout the same pentad for each year over the period of 2000–2020, to enable a comparison of relative SM changes throughout the same time each year. The 5th percentile was the difference between the percentiles at two adjacent pentads, which was calculated for the whole time series. To explore how fast flash drought evolves, we paid close attention to the onset development of flash droughts. As shown in Fig. 5, we calculated the decline rate of SM for the onset development period at each grid point. The onset point is defined as the first point where SM is over the 40th percentile and then the SM decline begins at a rate greater than the 5th percentile. The end point of the onset development is defined as the first point where SM drops below the 20th percentile. Therefore, the onset development phase of flash droughts starts from the onset point ($t_o$) and terminates at the end point ($t_e$). The period from $t_o$ to $t_e$ is the timescale of the onset development phase, which is defined as the onset time of flash droughts, suggesting a rapid onset of drying. The period from $t_e$ to $t_p$ refers to the duration under drought condition, and during such period the impacts may manifest and occur with a relatively long duration. To adequately reflect drought

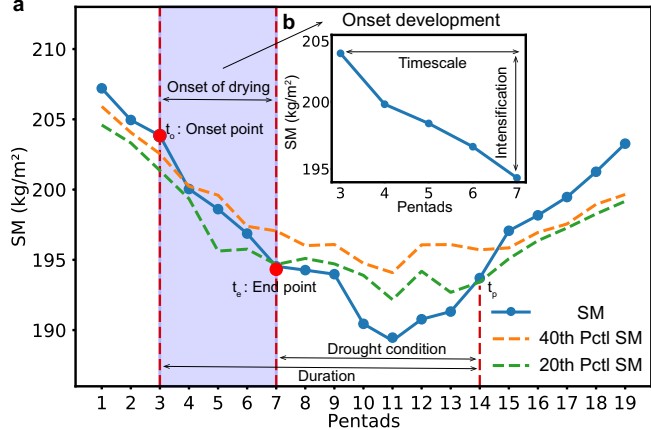

**Fig. 5 Schematic representation of the method used to identify a flash drought event. a** Schematic representation of the whole phase of a flash drought event. SM decreases from above the 40th percentile to below the 20th percentile with an average decline rate of no less than the 5th percentile for each pentad, and SM below the 20th percentile should last for no less than 3 pentads. The blue solid line represents the 5-day mean SM percentile for a grid point. The orange and green dashed lines represent the wet (the 40th percentile at a particular time of the year during the period 2000-2020) and the dry (the 20th percentiles at a particular time of the year during the period 2000-2020) conditions of SM, respectively. The purple shaded area represents the onset development of flash droughts. **b** Schematic representation of the onset phase of a flash drought event.

intensity and duration that can diminish crop productivity and yield, SM should not only drop below the 20th percentile but also last for at least three pentads ($t_p−t_e ≥ 3$). This criterion can also be used to exclude those events that decrease from above the 40th percentile to below the 20th percentile, and then recover up to the 40th percentile quickly (see Supplementary Fig. 19, a number of events experienced such a rapid decline in SM but recovered quickly, which should be excluded). Since the identification of flash droughts should take into account both flash (a rapid onset of drying, $t_o−t_e$) and drought severity (SM drops below a specific threshold for a period of time, $t_e−t_p$), the whole duration of flash droughts begins from $t_o$ to $t_p$. Specifically, the intensification rate and the drought condition are expressed in Eqs. (1) and (2), respectively.

$$\text{Intensification rate}: \begin{cases} \frac{SM(t_{i+1})-SM(t_i)}{t_{i+1}-t_i} \geq \text{5th percentile} \\ 0 < t_e - t_o \leq 5 \end{cases} \quad (1)$$

$$\text{Drought condition}: \begin{cases} SM(t_o) \geq \text{40th percentile} \\ SM(t_e) \leq \text{20th percentile} \\ t_p - t_e \geq 3 \end{cases} \quad (2)$$

**Detection of temporal trends**. The Mann–Kendall (M–K)[65,66] method is a nonparametric test, which is commonly used to examine whether there is a monotonic trend in the time series of the variable of interest. In the M–K test, the null hypothesis, $H_0$, is that there is no monotonic trend in the series. The alternative hypothesis, $H_1$, is that the data has a monotonic trend (positive or negative). Positive values of standardized test statistic $Z_{MK}$ indicate an increasing trend in the flash drought time series, whereas negative $Z_{MK}$ values suggest a decreasing trend. The advantages of this method are that statistical analysis is not required and samples are not required to follow a certain distribution. Thus, this method is not affected by a few abnormal values, and can be used to well characterize the trend of a time series. The M–K trend analysis was performed in this study to examine the trend of flash droughts on a global scale. For a given time series ($x_1, …, x_n$), the test statistic $Z_{MK}$ was calculated as follows:

$$S = \sum_{i=1}^{n-1} \sum_{j=i+1}^{n} \text{sign}(x_j - x_i) \quad (3)$$

$$\text{sign}(x_j - x_i) = \begin{cases} +1, x_j > x_i \\ 0, x_j = x_i \\ -1, x_j < x_i \end{cases} \quad (4)$$

$$\text{Var}(S) = \frac{1}{18}\left[n(n-1)(2n+5) - \sum_p t_p(t_p - 1)(2t_p + 5)\right] \quad (5)$$

$$Z_{MK} = \begin{cases} \frac{S-1}{\sqrt{\text{Var}(S)}} & \text{if } S > 0 \\ 0 & \text{if } S = 0 \\ \frac{S+1}{\sqrt{\text{Var}(S)}} & \text{if } S < 0 \end{cases} \quad (6)$$

where $n$ is the length of the time series. $x_i$ and $x_j$ are the sequential data in time series. $t_p$ is the number of ties of the $p$th value.

**Bivariate copulas and PMF.** The bivariate copulas are mathematical functions that can be used to describe the dependence between two random variables and to derive their joint distribution. The main advantage of copulas is its ability to overcome the shortcoming of assessing the co-occurrence rate of two climate extremes with few samples[67]. The joint distribution of random variables $X$ and $Y$ can be expressed as

$$F_{X,Y}(x, y) = P(X \le x, Y \le y) \quad (7)$$

where $X$ and $Y$ are random variables, and $P$ is their joint distribution. $F_X(x) = P(X \le x)$ and $F_Y(y) = P(Y \le y)$ are the marginal probability distribution of $X$ and $Y$, respectively. The joint cumulative distribution function (CDF) of $X$ and $Y$ can be expressed as

$$F_{X,Y}(x, y) = C[F_X(x), F_Y(y)] = C(u, v), 0 \le u, v \le 1 \quad (8)$$

where $F_X(x)$ and $F_Y(y)$ are transformed into two uniformly distributed random variables $u$ and $v$, and $C$ is a copula function. The copula families, including Gassian, Student't, Clayton, Gumbel, and Frank copula, were used to model the dependence structures of random variables. For each grid point, the optimal copula model was selected based on the Bayesian Information Criterion to well represent the dependence structure between two random variables.

In this study, we assessed the joint probabilities of extreme SM-VPD, SM-T, and SM-P at different thresholds. For example, to assess the joint probability of extreme SM (below the 30th percentile) and VPD (above the 90th percentile), the probability of such a compound event can be calculated as

$$p = P(u < 0.3 \cap v > 0.9) = P(u < 0.3) - P(u < 0.3 \cap v \le 0.9) = 0.3 - C(0.3, 0.9) \quad (9)$$

We also calculated the PMF to assess the increase in flash drought frequency due to the covariations between SM and VPD, SM and T, as well as SM and P, and a PMF value of 1 indicates that there is no change in frequency.

**Definition of dryland.** The global arid, transitional, and humid regimes can be identified as regions with an aridity index (AI)[68]. The AI, expressed as the ratio of P to PET, is a widely used indicator of regional moisture conditions, which is an effective index used to classify arid and humid zones. The interplay between water supply and demand, including both P and potential evaporation (Ep), is critical to the assessment of changes in dryness and dryland dynamics. The AI can thus be calculated based on the ratio between average annual P and Ep, which represents the characteristics of dryness/desertification over a specific region. In this study, we used monthly P and Ep, obtained from the CRU, to calculate the AI value.

$$AI = \frac{Ep}{P} \begin{cases} \text{arid}\,(AI > 2.25) \\ \text{transitional}\,(0.9 < AI \le 2.25) \\ \text{humid}\,(AI \le 0.9) \end{cases} \quad (10)$$

## Data availability

### SM data

Daily surface and root-zone SM were obtained from the Global Land Evaporation Amsterdam Model (GLEAM) (https://www.gleam.eu/) and two NASA GLDAS-2 (Global Land Data Assimilation System Version 2) models[69,70] including Noah[71–73] and Catchment land surface models (CLSM)[74] (https://disc.sci.gsfc.nasa.gov/datasets?keywords=GLDAS). The GLEAM SM datasets perform well against SM measurements from eddy covariance towers and SM sensors across a broad range of ecosystems[75]. GLDAS-2 has been proven to be able to reflect global and regional trends and patterns in SM[76,77]. Due to the good performance of GLDAS, it has been widely used to analyze global and regional SM changes[78,79]. Specifically, the root-zone SM, derived from each land surface model, was used to identify flash droughts since the rapid decline in SM can serve as a precursor for the occurrence of flash droughts, particularly when the plant-available SM approaches the wilting point. The daily GLEAM SM data at a spatial resolution of 0.25° × 0.25 were aggregated to the same resolution of GLDAS-2 at 1° × 1° and a temporal resolution of pentads for 2000−2020.

### Climatic data

We used daily climatic data from the Modern-Era Retrospective analysis for Research and Applications, Version 2 (MERRA-2) to calculate daily VPD. In MERRA-2, global T and humidity data are largely determined from the direct assimilation of satellite radiances (see https://gmao.gsfc.nasa.gov/pubs/docs/McCarty885.pdf). We used daily

near-surface (2-m) T and dew-point T data for 2000–2020. Daily VPD was calculated as the difference between saturated water vapor pressure, determined by near-surface T, and actual water vapor pressure, determined by dew-point T. The daily P, near-surface T, and VPD were used to access the drivers of the rapid onset of flash droughts. In addition, the monthly potential evaporation (Ep) and P from the Climatic Research Unit (CRU) (https://crudata.uea.ac.uk/cru/data/hrg/) were used to calculate AI for classifying arid/humid zones throughout the world. All these data were aggregated to the same resolution of SM data and a temporal resolution of pentads for 2000−2020. The source data for the figures are publicly available at: https://doi.org/10.5281/zenodo.5824138.

## Code availability

The drawing plots and computer codes are made using the open-source software R 4.0.2 and Python 3.8. The code that supports the findings of this study is publicly available at: https://doi.org/10.5281/zenodo.5824138.

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

## Acknowledgements

S.W. acknowledges support from the National Natural Science Foundation of China (grant no. 51809223) and the Hong Kong Research Grants Council Early Career Scheme (grant no. 25222319). The GLDAS datasets used in this study were acquired as part of the mission of NASA's Earth Science Division and distributed by the Goddard Earth Sciences (GES) Data and Information Services Center (DISC).

## Author contributions

S.W. conceived and supervised the study. Y.Q. and S.W. carried out the analysis and wrote the paper. Y.Q. and S.W. contributed equally to this work. B.C.A. and Z.-L.Y. provided comments and suggestions for improving the quality of this paper.

## Competing interests

The authors declare no competing interests.
