## [Peer Review File · Nature Communications]

Accelerating flash droughts induced by the joint influence of soil moisture depletion and atmospheric aridityReviewers' Comments:

Reviewer #1:

Remarks to the Author:

Please see attachment.

Comments on “More rapid intensification of flash droughts with shorter onset timescales”

(Manuscript # NCOMMS-21-07222)

by Y. Qing et al.

April 8, 2021

1 General Comments

This manuscript presents a study that investigate the intensification of flash droughts worldwide. The authors argue that there is a lack of a global assessment on the evolution and severity of flash droughts. To adress this deficit, the authors use the dataset GLDAS-2 with 0.25 x 0.25 degree resolution that contain SM estimated with Noah and VIC. The subject covered by this study is a highly relevant research topic having broad scientific and public interest, hence, it is highly suitable for this journal. In the present manuscript, however, there are many key points that have to be clarified before publication.

2 Specific Comments

- My major concerns with this manuscript are partly related with its novelty (many papers on the subject...) and the lack of a serious uncertainty analysis of the results. Considering that the variables derivedd from GLDAS-2 are based on two land surface model (Noah) and knowing that the uncertainty of Noah and VIC LSMs is large (see Cuntz. et al. JGR, Samaniego et al. BAMS 2019, among others), I consider that not estimating confidence

intervals for all the key statistics reported in this study constitute a crucial shortcoming that diminished the confidence of the study and does not qualify for publication in Nature journals.

- Another critical issue for me is the estimation of the SM percentile. Are percentiles estimated for every grid independently or for the whole field? Is the estimated SM deseasonized before the percentile is estimated? If not, a serious error is made. This is not clear for me at the moment. Moreover, why not to use a Soil Moisture Index as in Sheffield and Samaniego (see below) that follows an accepted and sound technique?
- In the list of indices, the Soil Moisture Index is missing. See
 - Sheffield, J., Goteti, G., Wen, F. & Wood, E. F. A simulated soil moisture based drought analysis for the United States. *Journal of Geophysical Research-Atmospheres* 109, 1–19 (2004).
 - Samaniego, L., Kumar, R. & Zink, M. Implications of Parameter Uncertainty on Soil Moisture Drought Analysis in Germany. *Journal of Hydrometeorology* 14, 47–68 (2013).
- Literature on flash droughts is missing (not exhaustively checked) but this one is missing, I guess.
 - Mo, K. C. & Lettenmaier, D. P. Heat wave flash droughts in decline. *Geophys. Res. Lett.* 42, 2823–2829 (2015).
- All estimates do not have uncertainty bounds. I consider this a must in this kind of studies. e.g., L 116 FF. (see above)
- What would happen to the conclusions if other LSMs are used? Would they still hold? This is the big issue with parametric uncertainty. In Samaniego et al JHM, for example, we showed that even with the same model, the conclusions are not conclusive and uncertainty must be taken into account. In Samaniego et al NCC, 2019, we showed that the LSMs used in this study (Noah-MP and VIC included) have large uncertainties, which also varies over space. This constitutes also a missing component of this study, i.e., how the uncertainty varies over hydro-climatic regions? I recommend the Authors to further investigate these issues.

- In eqs. 1,2, indicate how the 5th percentile is estimated.
- In eq. 6. indicate how the sample (x_1, \dots, x_n) is obtained. A single model?
- SMI compared with SM is preferable because it remove the bias that each LSM introduce into the SM. Each LSM has specific parameterizations that lead to great variations in soil parameters (see Samaniego et al, HESS 2017). If SMI is not used, but SM, i wonder how the authors could have used the data? For the bias between simulated SM see Fig 1. in
 - Wang, A., Bohn, T. J., Mahanama, S. P., Koster, R. D. & Lettenmaier, D. P. Multimodel Ensemble Reconstruction of Drought over the Continental United States. *Journal of Climate* 22, 2694–2712 (2009)

As a result, SMI is a must since it removes the bias and the seasonality. Koster et al also indicated in several studies that the SM anomalies should be used to avoid this issue. I recommend the authors to redo the analyses with this technique. The SMI code we use, for example, is open source <https://git.ufz.de/chs/progs/SMI>.

3 Editing Comments

This manuscript require also the following corrections:

1. The manuscript should be edited for Nature type journals. There are many templates for this purpose and even one in LaTeX. Nature papers do not have headings.
2. FAIR publishing (e.g., in AGU) does not contemplate this kind of statement “Codes will be made available upon request.” Codes must be in github or in a opne repository.
3. Conclusions are too succinct. An in-depth discussion is missing. What are the implications for other disciplines? Issues with uncertainty? Future steps? These issues should be discused in detail.

4 Final Remarks

Considering the major issues I described above, I consider that the Authors should rethink the message of the manuscript and redo the analysis as suggested. As I said before, the topic is very interesting, but I do not see much insights yet. But, perhaps I overlook something...

Based on the comments mentioned above and bearing in mind the NCOMMS publishing standards for a research article, I recommend **to reject this manuscript on its current form** and invite them to resubmit when the revision is concluded.

L. Samaniego

Reviewer #2:

Remarks to the Author:

NCOMMS-21-07222 Review of Qing et al.

Summary: The authors are looking at a global assessment of flash drought onset. Their findings suggest that flash droughts are showing trends toward more rapid intensification events over a shorter period of time. In addition, they found most flash droughts tend to occur in humid and semi-humid areas around the world. The author team brings attention to the need to upgrade drought prediction and early warning systems to better capture this breed of flash droughts.

Abstract:

The very first sentence seems like an overstatement and I don't see references/facts in the body of the paper to support it. What is this statement based on? There isn't a lot of in depth economic impact data that would allow for differentiating how flash droughts are causing "more" environmental or agricultural impacts as compared to the other types of drought. I don't know if I've seen any studies that actually quantify/confirm this statement. The fact that there hasn't been an adopted standard global definition of flash drought makes it very difficult to differentiate flash drought impacts from "normal" drought impacts. Long-term, multi-year droughts can have as much, if not more, impact as a quick onset drought due to compounding/cascading effects that occur over longer periods that aren't confined only to flash drought events. Later in the Introduction (Line 29-30), the authors contradict their impact statement by saying there is the "potential" to cause more severe impacts than slow developing droughts.

1. Introduction:

Lines 24-25: Timing of the onset isn't mentioned in regards to the flash drought of 2012 (Lines 31-33). In fact, the authors should state that they are referring to the 2012 drought if that is in fact the drought they are referring to here? The same onset during the dry season (winter) would have likely had very different effects vs. this flash drought occurring during critical stages of the crops growing season. This isn't factored in or mentioned?

Line 57: several definitions were proposed based on the US Drought Monitor. Are you referring to the classification (percentile/quantile) system?

Lines 72-79: Wouldn't the inclusion of oceanic oscillation/teleconnections strengthen this "influence" passage? Hard to include just atmospheric connection when the system is coupled between land/oceans. I understand that this isn't an attribution study, but the door was opened when you mentioned the atmospheric influence.

Line 100-103: I think there are a couple of other teams that have looked at flash drought globally, which wouldn't make this the "first" such study. This includes work by Koster et al. 2019 (DOI: 10.1175/JHM-D-18-0242.1), which looked at ET and soil moisture. I didn't see this Koster paper referenced or listed in the references. Some rephrasing of this sentence is needed to account for this as it is true that there aren't a lot of global studies out there in terms of flash drought that I'm aware of. Perhaps the emphasis should be on the onset/intensification angle as that would differentiate it more from the Koster method?

2. Results and Discussion:

Line 112: It is very hard to see any patterns given the small global maps in Figure 1. This is simply too many global maps on one figure. Even half of these examples may be too much for just one figure. Please enlarge the figure as it is unreadable for the most part.

Line 122: How do you account for droughts that manifest from rapid onset droughts to other types of drought such as agricultural or hydrological? The majority of flash drought studies (Lisonbee et al., 2021:

DOI: doi.org/10.46275/JOASC.2021.02.001) favor the idea that flash drought is not about the duration of the event. It is more about the rapid onset and intensification over a short period, not that flash droughts only last a short period of time. You mention this in Lines 128-130 so it is confusing to have the pentad limitation yet state that duration cannot guarantee it being a flash drought. It seems contradictory to me. Please note I realize that the Lisonbee reference would not have been available to you most likely until after you submitted this paper, but it is worth mentioning and taking account of

now.

Lines 132-134: Same comment as above dealing with duration. One pentad is very unlikely to result in any impacts and if there are no impacts is this really even a drought? Even at two pentads this would be rare and subject to extreme timing when it comes to impacts. Such a short time of dry days (1 or 2 pentads) would most likely need to be accompanied by very high temperatures and wind to have conditions that would lead to impacts). I believe you do go on to mention this later, but it should be mentioned now as well). This is a fundamental flaw with defining flash drought by duration.

Perhaps duration is a piece of the flash drought definition when it comes to onset, but not likely for duration as a drought may linger on after a rapid onset/intensification has taken place. You do seem to imply/focus on onset in Section 2.2, but I'm getting mixed messages in your methods/findings up to this point. The key phrasing here in Line 134 is "events that last for only one pentad."

Lines 191-92: Do you mean for "global" drought early warning indicators in regards to weekly/monthly updates? At the country level, a vast majority of indicators (although not for all countries) are available daily or weekly and at relatively high resolution (1-5km or better). The bigger issue is these triggers aren't linked to adequate planning or decision making, or are simply ignored given the stigma of drought being a "slow-onset" development hazard. We know that simply isn't true and this flash drought work is timely and relevant.

Lines 246-48: Again, I believe the tools are already there to develop the systems. Acceptance of rapid onset events and links to decision making and planning are the real gap. You are correct though in emphasizing the need to integrate these tools into our drought early warning systems. Sub-seasonal-to-seasonal (S2S) prediction needs in the context of drought are essential as well of course.

Line 260: Do you mean 20th percentile and not 2th, or do you mean 2nd?

Line 265: You should state clearly what your depth levels are, particularly the "top" layers vs. the "root zone". Most may have a general idea of what you mean by root zone (1m or so), but top could vary wildly depending on models and such. How many cm should be listed here and not just in a figure.

Line 297-98: Again, may want to tone down the "first time" language given Koster work.

Line 296: Seems a bit disjointed to read this and have the Conclusions come before both the Results and Discussion as well as the Methods? A lot of questions could be answered by putting the Methods earlier.

Lines 363-64: I agree with your wording here to exclude those events that recover quickly. Again, this goes towards not defining flash drought by duration.

FIGURES:

LN 660 Figure 1: Too many graphics in this figure. The world map views don't project well and it is hard to see colors/patterns...even in the defined red box regions on the maps. Same goes for the soil moisture plots...all of these need to be larger and/or broken out into two separate figures at a minimum. Even at 125% magnification, they aren't very readable.

LN 681 Figure 3: Same issue as figure 1, but not as bad given only 3 global map views. Still, not very readable except for the pentad bar charts. Please make them larger.

LN 688 Figure 4f: needs to be larger.

SUPPLEMENTARY MATERIAL

Figure 1 a-j: Not readable. Too small. Separate Figure 1 needed only showing a few world maps at a time so we can see the patterns better. Figure 10 is much better

Figure 8: Although only showing three global maps, they are still too small and hard to read even at 125% magnification. Need to make them bigger and perhaps only stack on top of each other instead of side-by-side? Figure 10 is readable as an example.

Figure 9: stack and make larger please...like Figure 10

Reviewer #3:

Remarks to the Author:

The premise of this study is interesting and, in my opinion, is worth exploring and potentially impactful. The question if flash droughts are intensifying more rapidly is of great interest for improving drought monitoring/early warning and reducing community vulnerability. However, there are numerous shortcomings and limitations in the present study that leave me wondering how robust and relevant the results and conclusions are for the stated purpose. Below is a list of my concerns, the largest of which is that the results of increase 3-pentad flash drought frequency are not linked to any physical change in regional climate, but are instead explained by vague references to changes in temperature and aridity. My recommendation is that the authors significantly revise the current study to focus on (1) quantifying the robustness of the observed trends and (2) analyzing the causes of the observed increase in 3-pentad flash drought events globally.

Specific Comments:

1) There are multiple false or misleading statements in this paper that are cause for concern, including:

1a) lines 35-38: At least for the references to the USA, neither of these studies pointed to climate change as a primary forcing of the 2012 or 2017 flash droughts, nor propose that flash droughts are increasing in frequency because of global warming. In fact, Hoerling et al. (2014) specifically list decadal variability as an important driver of the risk of 2012-like drought in the central U.S.

1b) lines 164-165: according to the U.S. Department of Agriculture, Alaska's agricultural product in 2006 (last survey) resulted in over \$60 million in cash receipts:

https://www.nass.usda.gov/Statistics_by_State/Alaska/About_Us/index.php. So, there is agriculture there and certainly in parts of the Sahara. Because this paper is not solely focused on flash drought impacts to agriculture, why exclude these regions because they are not agriculturally productive relative to other regions? Certainly a flash drought could have impacts on water resources or permafrost development in Alaska.

1c) lines 192-195: It looks from Figure 2 that the majority of flash droughts in each region occur in 3 pentads, not 1 or 2. But by grouping 1, 2, and 3 pentad developments together as you do here, it implies these events occur much more quickly. This is restated in lines 304-306. As most drought monitoring products are updated daily or weekly, I think they are capable of monitoring and maybe providing early warning for events that develop over a 15 day period.

1d) line 298: this really isn't a "new" definition, but mostly an adoption of a definition that's been around for 5-6 years now. Give credit to the definitions you adopted here.

2) The finding of increased frequency of 3-pentad development in place of 4- or 5-pentad development is interesting and noteworthy; however, the current study provides no satisfying explanations as to why this is happening at a global scale. Lines 132-160 (which I think could be removed to the benefit of the study) only describe a typical monsoonal climate and the atmospheric driver of essentially all warm season droughts in the central U.S. This discussion does not discriminate the factors at play specifically for flash drought in these regions, nor does it suppose why an increase in more rapid onset droughts has occurred over the last 20 years. Lines 221-232 provide some description of the meteorological conditions associated with flash drought, but again do not go far enough to explaining the reported increasing trends. This also brings into account the issue of no tests for statistical significance. The shaded time series plots in Figure 4 show quite a bit of variability. Could you at least report the calculated p-values to give the reader an idea of the confidence in the trend relative to the time series noise? This is particularly important given the relatively short study period.

3) Somewhat related to the last comment, without any notion of the physical causes of reported changes in flash drought intensification rate, it is difficult to assess where and how to improve operational drought monitoring and prediction. The statement on lines 310-313 and similar ones throughout the paper are quite vague and not really helpful to improving flash drought early warning. The majority of operational drought monitoring products are updated daily or weekly and can therefore capture a 3-, 4-, or 5-pentad flash drought. Can the authors demonstrate what is causing the reported increase in more rapid flash drought intensification and give some more relevant

information for improving drought monitoring and prediction?

4) Lines 12-13: Despite the argument that flash drought causes more impacts being used is numerous studies, it has actually never been explicitly studies or demonstrated (at least to my knowledge). Inarguably, flash drought provides less warning time for preparation, but I have not seen an objective evaluation of disproportionate impacts from the lack of preparation relative to a slower evolving drought. I don't think this study would be weakened by removing this statement.

Technical Edits:

1) Line 24: presumably, flash droughts have occurred prior to their scientific recognition. I suggest rephrasing to "a newly defined drought type..."

2) Line 29: "no early warning" should really be "less early warning". Also line 32: there was some early warning in 2012, just not as much as other droughts.

3) Line 49: "access" should be "assess"

4) Figure 1: why are the methods referred to as "new" and "old"? They were developed within 3-4 years of each other.

5) line 135: "mostly" should be "most"

6) Line 174 (and throughout): do you mean that the intensification rate was 28.6 percentiles? Because the 28.6th percentile refers to a specific percentile of a distribution, not how many percentiles a condition has increased or decreased.

7) Lines 178-180: there is some circular logic here. You defined flash droughts as those in which onset development occurs in 5 pentads or fewer, then find that these events onset more quickly than traditional droughts. Of course they do, that's how they were defined here.

8) What is the trend unit here? Is it flash drought events per year, per decade?

9) Line 331: can you provide more detail (perhaps in supplemental material) about the percentile calculation? Did you remove any seasonal cycle of soil moisture prior to calculating percentiles? This will be important in regions with pronounced seasonal drying and wetting.

10) Lines 359-364: the duration component is important to include, I'm glad that was done here. However, did you also include a duration component for flash drought recovery? For example, I can imagine a situation where the soil moisture declines to below the 20th percentile, remains there for 2-3 pentads, recovers quickly to the 40th percentile, and remains there for 1 pentad before quickly returning to the 20th or below. In that situation is there a single flash drought, or 2 flash droughts?

Black font = Reviewers' comments
Blue font = Response to reviewers' comments
Red font = Changes in the text of the manuscript

Responses to Reviewer #1:

General comments:

This manuscript presents a study that investigate the intensification of flash droughts worldwide. The authors argue that there is a lack of a global assessment on the evolution and severity of flash droughts. To address this deficit, the authors use the dataset GLDAS-2 with 0.25×0.25 degree resolution that contain SM estimated with Noah and VIC. The subject covered by this study is a highly relevant research topic having broad scientific and public interest, hence, it is highly suitable for this journal. In the present manuscript, however, there are many key points that have to be clarified before publication.

Response: We appreciate the reviewer's positive comment. To better explain our findings and conclusions, we have performed additional experiments with a more in-depth analysis and discussion, especially in terms of quantifying the robustness of conclusions and adding underlying physical mechanisms explaining our main findings. Our responses to the reviewer's specific comments are shown as follows.

Specific comments:

Comment #1) My major concerns with this manuscript are partly related with its novelty (many papers on the subject) and the lack of a serious uncertainty analysis of the results. Considering that the variables derived from GLDAS-2 are based on two land surface model (Noah) and knowing that the uncertainty of Noah and VIC LSMs is large (see Cuntz. et al. JGR, Samaniego et al. BAMS 2019, among others), I consider that not estimating confidence intervals for all the key statistics reported in this study constitute a crucial shortcoming that diminished the confidence of the study and does not qualify for publication in Nature journals.

Response: We appreciate the reviewer's constructive comment. We agree that uncertainty exists in the variables derived from GLDAS-2, based on two land surface models (Cuntz et al. 2016; Samaniego et al. 2019). To strengthen the reliability of the results and to reduce the uncertainty in LSMs, we have added the Global Land Evaporation Amsterdam Model (GLEAM) dataset that provides observationally constrained global daily surface and root-zone soil moisture (SM) spanning the period from 2000 to 2020. The GLEAM surface SM was produced by employing an improved SM data assimilation system using three independent SM datasets, including two satellite-based SM products from the European Space Agency (ESA) Climate Change Initiative (CCI) and Soil Moisture Ocean Salinity (SMOS) and the surface SM from the Noah model in the Global Land Data Assimilation System (GLDAS). The root-zone SM was modelled from a three-layer water balance module with the input of precipitation infiltration and the outputs of evapotranspiration and drainage.

In addition, we have compared all land surface models with GLDAS-2, including VIC, CLSM, and Noah. Based on the comparative analysis, we selected Noah and CLSM and removed VIC that performed poorly in comparison with the GLEAM dataset. Therefore, multiple data sources (GLEAM, Noah, and CLSM) were eventually used in this paper to reduce the

uncertainty in LSMs, and the confidence intervals for all the key statistics were then estimated according to the reviewer's comment. The detailed analysis of the selection of data sources has been added into the Supplementary Material. And we have revised relevant statements as follows.

“Daily surface and root-zone SM were obtained from the Global Land Evaporation Amsterdam Model (GLEAM) (<https://www.gleam.eu/>) and two NASA GLDAS-2 (Global Land Data Assimilation System Version 2) models^{69,70} including Noah^{71,72,73} and Catchment land surface models (CLSM)⁷⁴ (<https://disc.sci.gsfc.nasa.gov/datasets?keywords=GLDAS>). The GLEAM SM datasets perform well against SM measurements from eddy covariance towers and SM sensors across a broad range of ecosystems⁷⁵. GLDAS-2 has been proven to be able to reflect global and regional trends and patterns in SM^{76,77}. Due to the good performance of GLDAS, it has been widely used to analyze global and regional SM changes^{78,79}. Specifically, the root-zone SM, derived from each land surface model, was used to identify flash droughts since the rapid decline in SM can serve as a precursor for the occurrence of flash droughts, particularly when the plant-available SM approaches the wilting point. The daily GLEAM SM data at a spatial resolution of 0.25°×0.25 were aggregated to the same resolution of GLDAS-2 at 1°×1° and a temporal resolution of pentads for 2000–2020.”

References:

- Cuntz, M. et al. The impact of standard and hard-coded parameters on the hydrologic fluxes in the Noah-MP land surface model. *J. Geophys. Res. Atmos.* **121** (18), 10,676–10,700 (2016).
- Samaniego, L. et al. Hydrological forecasts and projections for improved decision-making in the water sector in Europe. *Bull. Am. Meteorol. Soc.* **100** (12), 2451–2472 (2019).

Comment #2) Another critical issue for me is the estimation of the SM percentile. Are percentiles estimated for every grid independently or for the whole field? Is the estimated SM deseasonized before the percentile is estimated? If not, a serious err is made. This is not clear for me at the moment. Moreover, why not to use a Soil Moisture Index as in Sheffield opr Samaniego (see below) that follows an accepted and sound technique?

Response: We regret for the unclear statement. Percentiles were determined for every grid independently throughout the calendar pentad of each year over the period of 2000 to 2020, to enable a comparison of relative SM change throughout the same time every year. Percentiles calculated based on the same pentad for years can avoid the influence of seasonal changes, which can thus avoid the effect of seasonal cycles on estimated SM. Fig. R1 shows the change of SM in a given grid when a flash drought event occurs. It can be seen that we used the percentile (the 20th and the 40th percentile SM) calculated based on the same calendar pentad of each year to assess the drought condition at the same time and place. In order to evaluate the decline rate of SM, we also calculated the percentile of SM based on the whole time series. Nevertheless, we did not use Soil Moisture Index (please refer to our response to *Comment #7* for the detailed comparison between our results and those calculated based on Soil Moisture

Index) because the pentad-drought condition was assessed based on soil moisture percentiles consistent with the (percentile/quantile) system of US Drought Monitor (USDM): conditions less (drier) than the 30th percentile, the 20th percentile, the 10th percentile, and the 5th percentile represent abnormally dry, moderate drought, severe drought, and extreme drought, respectively (Svoboda et al., 2002). The relevant statement has been revised as follows.

“It should be noted that the 40th and 20th percentiles were determined throughout the same pentad for each year over the period of 2000 to 2020, to enable a comparison of relative SM changes throughout the same time each year. The 5th percentile was the difference between the percentiles at two adjacent pentads, which was calculated for the whole time series.”

Fig. R1 Schematic representation of the method used to identify a flash drought event. a Schematic representation of the whole phase of a flash drought event. SM decreases from above the 40th percentile to below the 20th percentile with an average decline rate of no less than the 5th percentile for each pentad, and SM below the 20th percentile should last for no less than 3 pentads. The blue solid line represents the 5-day mean SM percentile for a grid point. The orange and green dashed lines represent the wet (the 40th percentile at a particular time of the year during the period 2000–2020) and the dry (the 20th percentiles at a particular time of the year during the period 2000–2020) conditions of SM, respectively. The purple shaded area represents the onset development of flash droughts. **b** Schematic representation of the onset phase of a flash drought event.

Reference:

Svoboda, M. et al. The Drought Monitor. *Bull. Am. Meteorol. Soc.* **83**, 1181–1190 (2002).

Comment #3) Literature on flash droughts is missing (not exhaustively checked) but this one is missing, I guess.

Response: We appreciate the reviewer's careful review, and have added this important reference in the revised version of the manuscript.

Reference:

Mo, K. C. & Lettenmaier, D. P. Heat wave flash droughts in decline. *Geophys. Res. Lett.* **42** (8), 2823–2829 (2015).

Comment #4) All estimates do not have uncertainty bounds. I consider this a must in this kind of studies. e.g., L 116 FF. (see above).

Response: We agree with the reviewer that the use of uncertainty bound is a must. Since multiple data sources are used to assess flash droughts, we have provided all estimates with uncertainty bounds (same as Fig. R2) in the revised version of the manuscript.

“Fig. 2 compares the percentages of flash droughts at different lead times relative to all flash droughts as well as their annual evolutions during the whole study period of 2000 to 2020. In general, 25.46–34.01% of flash droughts developed in 1 pentad, especially in Southeast Asia, South Asia, Western North America, and Central America. About 22.68–33.22% and 23.35–34.66% of flash droughts occurred in 2 and 3 pentads, respectively. In comparison, there are 7.82–18.14% and 1.58–3.04% of flash droughts developing in 4 and 5 pentads, respectively.”

“We find that the proportion of 1-pentad flash droughts in all flash drought events shows a significant ($P < 0.01$) increasing trend globally during the whole period 2000–2020 for all three datasets. The magnitude of estimated slope suggests an increase of 23.46–29.09% in the proportion of 1-pentad flash droughts during the study period for all three datasets, with an annual growth of 1.12–1.39% (Fig. 2c). Over most regions, the 1-pentad flash droughts also show an increasing trend, especially in Southern South America, Southeast Asia, Central North America, Southern Africa, and Eastern Africa where the increase in the proportion of 1-pentad flash droughts is up to 20% (Supplementary Fig. 5), indicating that more flash drought events develop within one week over these regions. By contrast, the proportions of flash droughts developing both in 2-3 and 4-5 pentads, derived from all three datasets, show a statistically significant ($P < 0.05$) decrease from 61.93–68.06% in 2000 to 53.05–57.51% in 2020, implying that flash droughts take less time to develop (Fig. 2d, e).”

Fig. R2 Comparison of percentages of flash droughts at different lead times relative to all flash droughts and their temporal evolutions. a The global proportion of flash droughts at different lead times relative to all flash droughts. The red dashed lines at the top of each bar represent the range of uncertainty in three different datasets. **b** The proportion of flash droughts at different lead times relative to all flash droughts over 21 regions. **c**, **d**, and **e** Temporal evolutions of the percentages of flash droughts with the lead times of 1-, 2-3, and 4-5 pentads relative to all flash droughts. The linear annual trends in the proportion of flash droughts at different lead times are estimated based on the Sen’s slope estimator, and statistical significances in trends are determined based on the MK test for the whole study period (2000–2020).

Comment #5) What would happen to the conclusions if other LSMs are used? Would they still hold? This is the big issue with parametric uncertainty. In Samaniego et al JHM, for example, we showed that even with the same model, the conclusions are not inclusive and uncertainty must be taken into account. In Samaniego et al NCC, 2019, we showed that the LSMs used in this study (Noah-MP and VIC included) have large uncertainties, which also varies over space. This constitute also a missing component of this study, i.e., how the uncertainty varies over hydro-climatic regions? I recommend the Authors to further investigate these issues.

Response: To address the reviewer's constructive comment, we have added the GLEAM dataset that provides observationally constrained global daily surface and root-zone SM. We have also compared all LSMs from GLDAS-2, including VIC, CLSM, and Noah, and then have removed VIC that performed poorly compared to the GLEAM dataset. To address the uncertainty in model parameters for different hydro-climatic regions, we realize that multiple data sources or multiple LSMs can be used as one of the effective ways to estimate the uncertainty bounds of the results (Mukherjee and Mishra, 2020; Yao et al. 2020; Hoffmann et al. 2020).

References:

- Mukherjee, S. & Mishra, A. K. Increase in compound drought and heatwaves in a warming world. *Geophys. Res. Lett.* e2020GL090617 (2020).
- Yao, N. et al. Projections of drought characteristics in China based on a standardized precipitation and evapotranspiration index and multiple GCMs. *Sci. Total Environ.* **704**, 135245 (2020).
- Hoffmann, D., Gallant, A. J. E. & Arblaster, J. M. Uncertainties in drought from index and data selection. *J. Geophys. Res. Atmos.* **125** (18), e2019JD031946 (2020).

Comment #6) In eqs. 1,2, indicate how the 5th percentile is estimated. In eq. 6. indicate how the sample (x_1, \dots, x_n) is obtained. A single model?

Response: We appreciate the reviewer's careful review. All the estimated percentiles and generated samples are based on a single model.

Comment #7) SMI compared with SM is preferable because it remove the bias that each LSM introduce into the SM. Each LSM has specific parameterizations that lead to great variations in soil parameters (see Samaniego et al, HESS 2017). If SMI is not used, but SM, i wonder how the authors could have used the data? For the bias between simulated SM see Fig 1. in. -Wang, A., Bohn, T. J., Mahanama, S. P., Koster, R. D. & Lettenmaier, D. P. Multimodel Ensemble Reconstruction of Drought over the Continental United States. *Journal of Climate* **22**, 2694–2712 (2009). As a result, SMI is a must since it removes the bias and the seasonality. Koster et al also indicated in several studies that the SM anomalies should be used to avoid this issue. I recommend the authors to redo the analyses with this technique. The SMI code we use, for example, is open source <https://git.ufz.de/chs/progs/SMI>.

Response: We appreciate the reviewer's comment. We have used multiple LSMs datasets and have also added a satellite observation-based dataset (GLEAM) to reduce the bias that each LSM introduces into the SM. Specifically, we have compared the annual total SM as well as the 40th- and 20th-percentile SM derived from GLEAM and GLDAS-2, and find that the pattern derived from the VIC model has a considerable difference with other three datasets, especially for the patterns of 40th- and 20th-percentile SM (Fig. R3). Thus, we have selected the SM data from GLEAM, Noah, and CLSM datasets. To further evaluate the performance of these three datasets used to capture flash drought events, we have compared flash drought frequencies identified by three different datasets. We find that even though the spatial patterns

of flash drought frequencies detected based on three datasets show some difference, hotspots are identified consistently and the difference in the total number of flash droughts identified based on three datasets lies between 0–7 (Fig. R4). Therefore, we have used these three datasets (GLEAM, Noah, and CLSM) to perform a robust assessment of flash droughts all over the world.

Fig. R3 Comparison of SM from four datasets. a–d Spatial patterns of annual total SM for GLEAM, Noah, CLSM, and VIC datasets. e–h Same as a–d, but for the 40th percentile SM. i–l Same as a–d, but for the 20th percentile SM.

Fig. R4 Comparison of frequencies of occurrence of flash droughts identified by GLEAM, Noah, and CLSM. a–c Spatial pattern of frequencies of occurrence of flash droughts identified

by GLEAM, Noah, and CLSM. **d** Boxplots of the total number of flash droughts identified by GLEAM, Noah, and CLSM.

According to the reviewer’s suggestion, we have redone the analyses using the Soil Moisture Index (SMI) (Samaniego et al. 2013; Samaniego et al. 2018). As the Standardized Soil Moisture Index (SSI) (Hao and AghaKouchak, 2013) is the most commonly used index, we have also compared it with the method (the percentile-based SM) we used in this paper. First, we have compared the traditional drought events captured by SSI, SMI, and the percentile-based SM since these indices are often used to quantify traditional drought events. As shown in Fig. R5, although the estimated frequencies of traditional droughts are different due to the different thresholds used by three indices (see Table R1), they show similar spatial patterns for all three datasets. Please note that the total root-zone saturated water content or porosity data related to three datasets we used are not available, and thus we used the SM from GLEAM and GLDAS to calculate the SMI, which might result in some deviations.

Table R1 Drought thresholds used by the percentile-based SM, SSI, and SMI.

Index	Percentile-based SM	SSI	SMI
Threshold	≤ 20 th percentile	≤ -1	≤ 0.2
Duration	≥ 3 pentads	≥ 3 pentads	≥ 3 pentads

Fig. R5 Comparison of frequencies of occurrence of traditional droughts identified by percentile-based SM, SSI, and SMI for GLEAM, Noah, and CLSM. a–c Spatial pattern of frequencies of occurrence of traditional droughts identified by percentile-based SM for GLEAM, Noah, and CLSM. **d–f** Spatial pattern of frequencies of occurrence of traditional droughts identified by SSI for GLEAM, Noah, and CLSM. **g–i** Spatial pattern of frequencies of occurrence of traditional droughts identified by SMI for GLEAM, Noah, and CLSM.

Second, we have compared the flash droughts identified by three indices (percentile-based SM,

SSI, and SMI). SSI and SMI are commonly used for analyzing drought events, but they cannot quantify the intensification rate of the flash drought onset. To solve this issue, we have calculated the SM percentile based on the whole time series and have used the difference between the percentiles at two adjacent pentads to represent the intensification rate. In other words, we have used SSI and SMI to capture drought events, and then have further identified the flash drought events (see Table R2) according to the intensification rate estimated based on the percentiles at two adjacent pentads. Fig. R6 shows the frequencies of flash droughts identified by the percentile-based SM, SSI, and SMI for three datasets. It can be seen that the spatial patterns of frequencies of flash droughts are similar. In particular, the hotspots captured by three definitions are highly consistent. It should be noted that the results from the percentile-based SM and SMI are similar both in terms of spatial distributions and the total number of flash drought events detected. In comparison, the total number of flash droughts captured by SSI is much lower due to the selected threshold. When the threshold increases, nonetheless, more flash drought events would be captured.

Table R2 Flash drought thresholds used by the percentile-based SM, SSI, and SMI.

Index	Percentile-based SM	SSI	SMI
Onset start	≥ 40 th percentile	≥ -0.5	≥ 0.4
Onset end	≤ 20 th percentile	≤ -1.5	≤ 0.2
Intensification rate	≥ 5 th percentile	≥ 5 th percentile	≥ 5 th percentile
Duration	≥ 3 pentads	≥ 3 pentads	≥ 3 pentads

Fig. R6 Comparison of frequencies of occurrence of flash droughts identified by the percentile-based SM, SSI, and SMI for GLEAM, Noah, and CLSM. a–c Spatial pattern of frequencies of occurrence of flash droughts identified by the percentile-based SM for GLEAM, Noah, and CLSM. **d–f** Spatial pattern of frequencies of occurrence of flash droughts identified by SSI for GLEAM, Noah, and CLSM. **g–i** Spatial pattern of frequencies of occurrence of flash droughts identified by SMI for GLEAM, Noah, and CLSM.

Third, we have also compared temporal evolutions of the percentages of flash droughts with the lead times of 1-, 2-3, and 4-5 pentads relative to all flash droughts identified by SMI and SSI. According to the percentile-based SM definition, the proportion of 1-pentad flash droughts in all flash drought events shows a significant ($P < 0.01$) increasing trend globally during the whole period 2000–2020 for all three datasets. The magnitude of estimated slope suggests an increase of 23.46–29.09% in the proportion of 1-pentad flash droughts during the study period for all three datasets, with an annual growth of 1.12–1.39%. However, the proportions of flash droughts developing in 4-5 pentads, derived from all three datasets, show a statistically significant ($P < 0.05$) decrease from 16.14–27.83% in 2000 to 4.89–10.48% in 2020, implying that flash droughts take less time to develop. As for SMI (Fig. R7a-c) and SSI (Fig. R7d-f), the proportions of 1-pentad flash droughts in all flash drought events show an increasing trend globally during the whole period 2000–2020 for all three datasets, while the proportions of flash droughts developing in 4-5 pentads show a statistically significant ($P < 0.05$) decrease trend. The proportions of flash droughts developing in 2-3 pentads for three datasets did not all pass the significance test. Consequently, the main conclusion is that the onset of flash droughts is getting faster, which is consistent with each other based on three different indices (the percentile-based SM, SSI, and SMI).

Fig. R7 Temporal evolution of the percentages of flash droughts with the lead times of 1-, 2-3, and 4-5 pentads relative to all flash droughts identified by SMI (a-c) and SSI (d-f).

Finally, according to our definition (the percentile-based SM), the 20th percentile threshold is selected as recommended by the USDM (US Drought Monitor) to represent the “Moderate Drought – D1” condition, under which vegetation may start showing signs of water stress. The minimum intensification period of 1 month (5 pentads) is consistent with the recommendation from Otkin et al. (2018). And the 5th percentile is the difference between the percentiles at two adjacent pentads, which is calculated for assessing the decline rate of SM. As all these criteria are based on the percentile calculation, we think the use of the percentile-based SM definition may better fit with the percentile system. Moreover, the lack of related data (the total root-zone

saturated water content or porosity data) makes it difficult to use the SMI. And some SMI results were also calculated based on the SM percentile, including SMPD (Ford and Labosier, 2017), SMVI (Osman et al. 2021), HWD (Mo and Lettenmaier, 2015, 2016), and UDSM.

References:

- Hao, Z. & AghaKouchak, A. Multivariate standardized drought index: a parametric multi-index model. *Adv. Water Res.* **57**, 12–18 (2013).
- Samaniego, L., Kumar, R. & Zink, M. Implications of parameter uncertainty on soil moisture drought analysis in Germany. *J. Hydrometeorol.* **14** (1), 47–68 (2013).
- Samaniego, L. et al. Anthropogenic warming exacerbates European soil moisture droughts[J]. *Nat. Clim. Chang.* **8** (5), 421–426 (2018).
- Ford, T. W. & Labosier, C. F. Meteorological conditions associated with the onset of flash drought in the Eastern United States. *Agric. For. Meteorol.* **247**, 414–423 (2017).
- Osman, M. et al. Flash drought onset over the Contiguous United States: Sensitivity of inventories and trends to quantitative definitions. *Hydrol. Earth Syst. Sci.* **25** (2), 565–581 (2021).
- Mo, K. C. & Lettenmaier, D. P. Heat wave flash droughts in decline. *Geophys. Res. Lett.* **42** (8), 2823–2829 (2015).
- Mo, K. C. & Lettenmaier, D. P. Precipitation deficit flash droughts over the United States. *J. Hydrometeorol.* **17** (4), 1169–1184 (2016).

Editing comments:

Comment #8) The manuscript should be edited for Nature type journals. There are many templates for this purpose and even one in LaTeX. Nature papers do not have headings.

Response: According to the reviewer’s comment, we have edited the manuscript for Nature type journals.

Comment #9) FAIR publishing (e.g., in AGU) does not contemplate this kind of statement “Codes will be made available upon request.” Codes must be in github or in an open repository.

Response: According to the reviewer’s comment, we have added the following statement in the revised version of the manuscript.

“The analysis code that supports the findings of this study is available at <https://github.com/QINGYAMIN/Code-flash-drought>.”

Comment #10) Conclusions are too succinct. An in-depth discussion is missing. What are the implications for other disciplines? Issues with uncertainty? Future steps? These issues should be discussed in detail.

Response: According to the reviewer’s constructive comment, we have conducted tests for

statistical significance to quantify the robustness of the observed trends (please refer to our response to *Comment #4*). Additionally, we have made great efforts to reveal the physical processes causing flash droughts at a regional to global scale. To address the reviewer's comment, we have added the following discussions in the revised version of the manuscript.

Attribution analysis on a regional scale

“To explore flash drought evolution characteristics and to conduct attribution analysis, we plotted the time series of anomalies of annual flash drought-affected land areas, annual mean T, annual total P, annual mean VPD, and relative SM change over the NEU-MEU (the total area of Northern Europe and Mediterranean Basin) during 2000–2020 (Fig. 3). These two places cover most of the European region, and the annual frequencies of flash droughts of Northern Europe and Mediterranean Basin show an increasing trend (Supplementary Fig. 9). It is found that high T, low P, and high VPD often coincide with a high percentage of flash drought-affected land areas. The high T, low P, and high VPD in 2007, 2014, 2016, and 2018 rank in the top 5 during 2000–2020 (Fig. 3a-d), and the percentage of land areas in 2007, 2014, 2016, and 2018 also rank in the top 5 (red boxes in Fig. 3). On the other hand, there are few significant correlations between T and the percentage of flash drought-affected land areas. The high percentage of flash drought areas is not always accompanied by high T, but it is always under the P-deficit and VPD condition (black box in Fig. 3). This suggests that the flash drought occurrence is not dominated by T. The high T can cause SM to drop, but such a decrease in SM anomalies is not large¹⁴. The low P and high VPD before or during the onset of flash drought events are needed to increase the SM deficits and to make the conditions more favorable for the occurrence of flash droughts. In this sense, P and VPD play an important role in enhancing the possibility of flash drought occurrence. In addition, the variation of annual mean VPD is similar to the variation of relative change in annual mean SM. Specifically, the relative change in SM increases (decreases) when VPD increases (decreases), suggesting that the increasing VPD triggers a rapid decline in SM. And the rapid decline in SM induces the occurrence of a flash drought event. Thus, high T, low P, and high VPD create a flash drought-prone environment, and VPD is a significant factor influencing the speed of onset, which tends to drive a rapid decline in SM to cause flash drought events.”

Fig. 3 Interannual variations of flash drought-affected land area and its related variables over the NEU-MEU from 2000 to 2020. a, e Anomalies of flash drought-affected land area and relative SM change over the NEU-MEU from three different datasets. The relative SM change is defined as the change of SM percentiles between adjacent pentads (the SM percentile at the current pentad minus the SM percentile at the next pentad), where positive values represent a decline in SM at the current pentad. **b, c, and d** Anomalies of annual mean T, annual total P, and annual mean VPD over the NEU-MEU.

Attribution analysis on a global scale

“There is a strong negative correlation between SM and VPD, and the low SM is always accompanied by the high VPD⁶⁰. In turn, the high VPD enhances the atmospheric evaporative demand, which can exacerbate the decline in SM through land–atmosphere interactions⁶¹. To investigate the contribution of VPD to the occurrence of flash droughts, we examined the underlying relationship between SM and VPD (Fig. 4). We find that SM is strongly correlated with VPD, which is evident in the bimodal distribution of SM and VPD: high VPD and low SM, and low VPD and high SM (Fig. 4b). To further explore the onset phase of flash droughts, we defined the bivariate extremes as those with SM below the 30th percentile (the 30th percentile was used to represent the mean condition of flash drought onset phase) and VPD above the 90th percentile. Probability multiplication factor (PMF) was calculated to assess the increasing frequency of compound events compared to those expected when SM and VPD were independent, and a PMF of 1 represents no increase in the joint probability. We find that certain regions with a high value of PMF are consistent with those experiencing a high frequency of flash drought events (dashed boxes in Fig. 1c and Fig. 4a). And even if the thresholds of SM and VPD change, similar patterns can also be obtained (Supplementary Fig. 10). We also calculated the PMF of SM-T and SM-P, and the corresponding patterns show less correlation with those of flash drought frequency compared with SM-VPD (Supplementary Fig. 11). More importantly, a positive correlation ($R=0.64$, the point of Tibet was excluded from correlation analysis) between the frequency of flash droughts occurring in 21 regions and PMF is found (Fig. 4c), especially for the Southeast Asia, Amazon Basin, and Southern South America, and Eastern North America where the joint probability of bivariate extremes is approximately 1.84–2.55 times higher than the probability of the independent combination. This indicates that the strong coupling of SM and VPD through land-atmosphere feedbacks contributes to the rapid decline in SM. Moreover, the positive correlation between the frequency of flash droughts occurring in 21 regions and PMF will increase to 0.75 when the threshold of extreme SM decreases to the 20th percentile, suggesting that the intensity of SM-VPD coupling will increase when there is a further decline in SM (Fig. 4d).”

Fig. 4 Relationship between SM and VPD. **a** Mean PMF of concurrent extreme SM (below the 30th percentile) and VPD (above the 90th percentile) for the period 2000–2020 based on three datasets. **b** Mean probability of percentile bins of SM and VPD across all land grid points. **c** Spatial relationship between the frequency of occurrence of flash droughts and the PMF of SM and VPD extremes (below the 30th percentile SM and above the 90th percentile VPD). **d** Spatial relationship between the frequency of occurrence of flash droughts and the PMF of SM and VPD extremes (below the 20th percentile SM and above the 90th percentile VPD). Each point in **c** and **d** is the mean of the results from three datasets.

Societal implications

“Collectively, our findings suggest that flash droughts tend to occur in humid and semi-humid regions, including Southeast Asia, East Asia, Amazon Basin, Eastern North America, and Southern South American. The identification of flash drought-prone regions and global hot spots can provide valuable insights to help inform policymakers and stakeholders on potential risks of flash droughts. Specifically, the onset development of flash droughts is becoming faster, approximately 25.46–34.01% of flash droughts occur within 5 days for the period 2000–2020, which poses a great challenge for drought monitoring. There are several products and indicators used to capture flash drought events, including the U.S. Drought Monitor (USDM), the Drought Impact Reporter (DIR), the U.S. Crops and Livestock in Drought, the Climate Prediction Center (CPC) SM, and the Vegetation Drought Response Index (VegDRI). Although some of them are available on a daily or weekly basis, but not for all countries. Thus, assessing the relatively fast onset development timescales of flash droughts can provide useful information for upgrading drought monitoring systems and indicators all over the world. To further improve the ability of monitoring and predicting flash droughts,

the criterion of a rapid intensification rate should be taken into account in addition to the relatively short onset timescales for capturing the unique characteristics of flash droughts (rapid onset and rapid intensification). The change in the state of the climate should also be incorporated into flash drought monitoring and prediction so that it remains meaningful in a warming climate.

It should also pay close attention to relevant compound and atmospheric circulation dynamics that may trigger the onset of flash droughts. Identifying the drivers of flash droughts and the associated factors that may speed up the rapid decline in SM is crucial to developing plausible risk mitigation scenarios based on multi-criteria analysis of potential weather conditions in different geographical contexts. According to our study, the atmospheric aridity (high T, low P, and high VPD) is conducive to flash drought development, and the resulting anomalies are likely to trigger the occurrence of flash droughts. Specifically, VPD is a significant factor causing the rapid decline in SM, and the SM-VPD coupling can further enhance the SM depletion. Thus, we should pay more attention to the regions with relatively high risk of compound extreme events, especially for compound drought and heatwave events. In addition, we should focus on the regions with strong land–atmosphere coupling since there is a high correlation between the joint probability of bivariate extremes and the frequency of flash droughts. To advance the understanding of flash droughts, more research efforts should be made, especially for quantifying environmental, social and economic vulnerability to flash droughts and their impacts.”

Future steps

“Flash droughts are characterized by rapid onset, and we find that the rapid onset can be attributed to large-scale atmospheric dynamics and land-atmosphere feedbacks. The local moisture imbalance is influenced by large-scale atmospheric dynamics through changing the local T, P, and VPD. The anomalies of multiple factors (T, P, and VPD) that occur simultaneously may trigger the rapid onset of flash droughts, such as the aforementioned case of NEU-MEU. In addition, we find that the strong coupling of SM and VPD also contributes to the development of flash droughts since the dependence between extreme high VPD and low SM leads to a rapid SM depletion and thus triggers flash drought development. Furthermore, the likelihood of the accelerating onset of flash droughts increases significantly due to anthropogenic climate change. For example, Wang and Yuan⁶⁴ found that anthropogenic climate change increased the likelihood of onset speed and intensity of the 2019-like flash drought event by $24 \pm 16\%$ and $37 \pm 9\%$, respectively. Moreover, the teleconnected ocean-atmosphere may influence the development of flash droughts through atmospheric circulation associated with a rapid decline in SM in conjunction with P deficit¹⁴. Vegetation is also a key component affecting the rapid onset development of flash droughts because of its important role in mediating the transpiration. Crops can suffer from moisture stress more quickly by pumping water from deep soil, which may trigger a rapid development of flash droughts. Exploring mechanisms of flash droughts would be challenging but a promising direction for unraveling the mystery of flash drought onset.”

Responses to Reviewer #2:

General comments:

The authors are looking at a global assessment of flash drought onset. Their findings suggest that flash droughts are showing trends toward more rapid intensification events over a shorter period of time. In addition, they found most flash droughts tend to occur in humid and semi-humid areas around the world. The author team brings attention to the need to upgrade drought prediction and early warning systems to better capture this breed of flash droughts.

Response: We appreciate the reviewer providing constructive comments and suggestions for improving the quality of our manuscript. Our responses to the reviewer's specific comments are shown as follows.

Specific comments:

Comment #1: Abstract) The very first sentence seems like an overstatement and I don't see references/facts in the body of the paper to support it. What is this statement based on? There isn't a lot of in-depth economic impact data that would allow for differentiating how flash droughts are causing "more" environmental or agricultural impacts as compared to the other types of drought. I don't know if I've seen any studies that actually quantify/confirm this statement. The fact that there hasn't been an adopted standard global definition of flash drought makes it very difficult to differentiate flash drought impacts from "normal" drought impacts. Long-term, multi-year droughts can have as much, if not more, impact as a quick onset drought due to compounding/cascading effects that occur over longer periods that aren't confined only to flash drought events. Later in the Introduction (Line 29-30), the authors contradict their impact statement by saying there is the "potential" to cause more severe impacts than slow developing droughts.

Response: We regret for the overstatement on the economic impact of flash droughts, and agree that there is no study actually confirming that flash droughts are causing more environmental or agricultural impacts as compared to the other types of drought. Thus, we have corrected this statement as follows.

"The emergence of flash drought has attracted widespread attention due to its rapid onset and devastating impacts on the environment and agriculture. However, little is known about the recent evolution of flash droughts in terms of the speed of onset and the causes of such a rapid onset phase of flash droughts."

Comment #2: Lines 24-25) Timing of the onset isn't mentioned in regards to the flash drought of 2012 (Lines 31-33). In fact, the authors should state that they are referring to the 2012 drought if that is in fact the drought they are referring to here? The same onset during the dry season (winter) would have likely had very different effects vs. this flash drought occurring during critical stages of the crops growing season. This isn't factored in or mentioned?

Response: We regret for the unclear statement. According to the reviewer's comment, we have

revised the statement as follows.

“For example, the 2012 flash drought that occurred over the Central United States caused a tremendous impact on agricultural production and the economy, with \$12 billion losses attributed to this event⁵. Furthermore, analysis of retrospective climate simulations found virtually no dry signal, owing to the unusual speed and intensity (developed in May and reached peak intensity by August), over the major corn-producing regions of the eastern Great Plains where this drought event resulted in major curtailment of corn crop yields.”

Reference:

Hoerling, M. P. et al. Causes and predictability of the 2012 Great Plains drought. *Bull. Am. Meteor. Soc.* **95**, 269–282 (2014).

Comment #3: Line 57) Several definitions were proposed based on the US Drought Monitor. Are you referring to the classification (percentile/quantile) system?

Response: We regret for the unclear statement, and have rewritten the sentence for better clarification.

“In addition, there are a variety of indices available to identify flash droughts, including the Evaporative Demand Drought Index (EDDI)^{17,18}, the Evaporative Stress Index (ESI)^{19,20}, the Standard Evaporative Stress Ratio (SESR)²¹, the combination of the ESI and the Rapid Change Rate Index (RCI)^{22,23,24}, P²⁵, vegetation²⁶, and SM^{9,27,28,29}. Several definitions were also proposed based on the classification (percentile/quantile) system of US Drought Monitor (USDM)^{30,31,32}.”

Comment #4: Lines 72-79) Wouldn't the inclusion of oceanic oscillation/teleconnections strengthen this “influence” passage? Hard to include just atmospheric connection when the system is coupled between land/oceans. I understand that this isn't an attribution study, but the door was opened when you mentioned the atmospheric influence.

Response: We appreciate the reviewer's insightful comment. The oceanic and/or atmospheric oscillations/teleconnections are important factors that may influence the occurrence of droughts, and we agree with the reviewer that atmospheric connection should be considered together with the coupling of land and oceans. According to the reviewer's comment, we have performed additional experiments to better explain our main findings. We find that there is no significant relationship between the rapid onset of flash droughts and oceanic oscillations/teleconnections, but the rapid onset can be attributed to large-scale atmospheric dynamics and land-atmosphere feedbacks. The local moisture imbalance is influenced by large-scale atmospheric dynamics through changing the local T, P, and VPD. The anomalies of multiple factors (T, P, and VPD) that occur simultaneously may trigger the rapid onset of flash droughts. In addition, we find that the strong coupling of SM and VPD also contributes to the development of flash droughts since the dependence between extreme high VPD and low SM leads to a rapid SM depletion and thus triggers flash drought development. We have added the

following discussions to explain the causes of the more rapid onset of flash drought events globally in the revised version of the manuscript.

“To explore flash drought evolution characteristics and to conduct attribution analysis, we plotted the time series of anomalies of annual flash drought-affected land areas, annual mean T, annual total P, annual mean VPD, and relative SM change over the NEU-MEU (the total area of Northern Europe and Mediterranean Basin) during 2000–2020 (Fig. 3). These two places cover most of the European region, and the annual frequencies of flash droughts of Northern Europe and Mediterranean Basin show an increasing trend (Supplementary Fig. 9). It is found that high T, low P, and high VPD often coincide with a high percentage of flash drought-affected land areas. The high T, low P, and high VPD in 2007, 2014, 2016, and 2018 rank in the top 5 during 2000–2020 (Fig. 3a-d), and the percentage of land areas in 2007, 2014, 2016, and 2018 also rank in the top 5 (red boxes in Fig. 3). On the other hand, there are few significant correlations between T and the percentage of flash drought-affected land areas. The high percentage of flash drought areas is not always accompanied by high T, but it is always under the P-deficit and VPD condition (black box in Fig. 3). This suggests that the flash drought occurrence is not dominated by T. The high T can cause SM to drop, but such a decrease in SM anomalies is not large¹⁴. The low P and high VPD before or during the onset of flash drought events are needed to increase the SM deficits and to make the conditions more favorable for the occurrence of flash droughts. In this sense, P and VPD play an important role in enhancing the possibility of flash drought occurrence. In addition, the variation of annual mean VPD is similar to the variation of relative change in annual mean SM. Specifically, the relative change in SM increases (decreases) when VPD increases (decreases), suggesting that the increasing VPD triggers a rapid decline in SM. And the rapid decline in SM induces the occurrence of a flash drought event. Thus, high T, low P, and high VPD create a flash drought-prone environment, and VPD is a significant factor influencing the speed of onset, which tends to drive a rapid decline in SM to cause flash drought events.

There is a strong negative correlation between SM and VPD, and the low SM is always accompanied by the high VPD⁶⁰. In turn, the high VPD enhances the atmospheric evaporative demand, which can exacerbate the decline in SM through land–atmosphere interactions⁶¹. To investigate the contribution of VPD to the occurrence of flash droughts, we examined the underlying relationship between SM and VPD (Fig. 4). We find that SM is strongly correlated with VPD, which is evident in the bimodal distribution of SM and VPD: high VPD and low SM, and low VPD and high SM (Fig. 4b). To further explore the onset phase of flash droughts, we defined the bivariate extremes as those with SM below the 30th percentile (the 30th percentile was used to represent the mean condition of flash drought onset phase) and VPD above the 90th percentile. Probability multiplication factor (PMF) was calculated to assess the increasing frequency of compound events compared to those expected when SM and VPD were independent, and a PMF of 1 represents no increase in the joint probability. We find that certain regions with a high value of PMF are consistent with those experiencing a high frequency of flash drought events (dashed boxes in Fig. 1c and Fig. 4a). And even if the thresholds of SM and VPD change, similar patterns

can also be obtained (Supplementary Fig. 10). We also calculated the PMF of SM-T and SM-P, and the corresponding patterns show less correlation with those of flash drought frequency compared with SM-VPD (Supplementary Fig. 11). More importantly, a positive correlation ($R=0.64$, the point of Tibet was excluded from correlation analysis) between the frequency of flash droughts occurring in 21 regions and PMF is found (Fig. 4c), especially for the Southeast Asia, Amazon Basin, and Southern South America, and Eastern North America where the joint probability of bivariate extremes is approximately 1.84–2.55 times higher than the probability of the independent combination. This indicates that the strong coupling of SM and VPD through land-atmosphere feedbacks contributes to the rapid decline in SM. Moreover, the positive correlation between the frequency of flash droughts occurring in 21 regions and PMF will increase to 0.75 when the threshold of extreme SM decreases to the 20th percentile, suggesting that the intensity of SM-VPD coupling will increase when there is a further decline in SM (Fig. 4d).”

Fig. 3 Interannual variations of flash drought-affected land area and its related variables over the NEU-MEU from 2000 to 2020. a, e Anomalies of flash drought-affected land area and relative SM change over the NEU-MEU from three different datasets. The relative SM change is defined as the change of SM percentiles between adjacent pentads (the SM percentile at the current pentad minus the SM percentile at the next pentad), where positive values represent a decline in SM at the current pentad. **b, c, and d** Anomalies of annual mean T, annual total P, and annual mean VPD over the NEU-MEU.

Fig. 4 Relationship between SM and VPD. a Mean PMF of concurrent extreme SM (below the 30th percentile) and VPD (above the 90th percentile) for the period 2000–2020 based on three datasets. **b** Mean probability of percentile bins of SM and VPD across all land grid points. **c** Spatial relationship between the frequency of occurrence of flash droughts and the PMF of SM and VPD extremes (below the 30th percentile SM and above the 90th percentile VPD). **d** Spatial relationship between the frequency of occurrence of flash droughts and the PMF of SM and VPD extremes (below the 20th percentile SM and above the 90th percentile VPD). Each point in **c** and **d** is the mean of the results from three datasets.

Comment #5: Lines 100-103) I think there are a couple of other teams that have looked at flash drought globally, which wouldn't make this the "first" such study. This includes work by Koster et al. 2019 (DOI: 10.1175/JHM-D-18-0242.1), which looked at ET and soil moisture. I didn't see this Koster paper referenced or listed in the references. Some rephrasing of this sentence is needed to account for this as it is true that there aren't a lot of global studies out there in terms of flash drought that I'm aware of. Perhaps the emphasis should be on the onset/intensification angle as that would differentiate it more from the Koster method?

Response: We appreciate the reviewer's careful review, and have added the Koster's paper in the list of references. According to the reviewer's suggestion, we have put emphasis on the rapid onset phase of flash droughts and underlying causes of a rapid onset in the revised version of the manuscript.

"A previous study produced a global map of flash droughts, with a focus on the contribution of evapotranspiration (ET) and P to the occurrence of flash

droughts⁴⁰. Nonetheless, few studies explore the rapid onset phase of flash droughts and underlying causes of the rapid onset which is the most important characteristic of flash droughts.”

Reference:

Koster, R. D., Schubert, S. D., Wang, H., Mahanama, S. P. & DeAngelis, A. M. Flash drought as captured by reanalysis data: Disentangling the contributions of precipitation deficit and excess evapotranspiration. *J. Hydrometeorol.* **20** (6), 1241–1258 (2019).

Comment #6: Line 112) It is very hard to see any patterns given the small global maps in Figure 1. This is simply too many global maps on one figure. Even half of these examples may be too much for just one figure. Please enlarge the figure as it is unreadable for the most part.

Response: According to all reviewers’ comments, we have updated the majority of the contents and associated figures in the revised version of the manuscript. Thus, all figures have been redrawn for improving the readability.

Comment #7: Line 122) How do you account for droughts that manifest from rapid onset droughts to other types of drought such as agricultural or hydrological? The majority of flash drought studies (Lisonbee et al., 2021;DOI: doi.org/10.46275/JOASC.2021.02.001) favor the idea that flash drought is not about the duration of the event. It is more about the rapid onset and intensification over a short period, not that flash droughts only last a short period of time. You mention this in Lines 128-130 so it is confusing to have the pentad limitation yet state that duration cannot guarantee it being a flash drought. It seems contradictory to me. Please note I realize that the Lisonbee reference would not have been available to you most likely until after you submitted this paper, but it is worth mentioning and taking account of now.

Response: We appreciate the reviewer’s constructive comment. It is an interesting topic on the evolution from rapid onset droughts to other types of droughts. For example, a precipitation deficiency may result in a rapid depletion of soil moisture, and such a rapid drought may have a further influence on surface or subsurface water supply (i.e., streamflow, reservoir and lake levels, groundwater) and cause the occurrence of hydrological drought. However, we focus on the rapid onset phase of flash droughts in this paper. Future work will be undertaken to detect the events that manifest from rapid onset droughts to other types of droughts including agricultural and hydrological droughts.

In addition, we regret for the unclear statement in Lines 128-130. As flash drought has a sudden onset and rapid intensification over a short period of time, the definition should consider not only a rapid onset but also a rapid rate of intensification. Duration is only used to identify flash droughts with the whole development time of only 5 days which are insufficient to diminish crop productivity and yield (Fig. R8b). Moreover, the only use of duration cannot guarantee that the drought events experience a rapid onset phase which is the most important characteristic of flash droughts. Consequently, the pentad limitation is applied to the whole development phase of flash drought, which is used to meet the requirement that the detected

flash drought can result in an adverse impact on the crop productivity and yield.

Fig. R8 Variation of SM percentiles for flash droughts of all grid points. a Variation of SM percentiles for flash droughts of all grid points, identified by intensification rate. **b** Variation of SM percentiles for flash droughts of all grid points, identified by duration. The color lines represent the mean of all flash drought detected from three models.

Reference:

Lisonbee, J., Woloszyn, M. & Skumanich, M. Making sense of flash drought: Definitions, indicators, and where we go from here. *J. Appl. Serv. Climatol.* 1–19 (2021).

Comment #8: Lines 132-134) Same comment as above dealing with duration. One pentad is very unlikely to result in any impacts and if there are no impacts is this really even a drought? Even at two pentads this would be rare and subject to extreme timing when it comes to impacts. Such a short time of dry days (1 or 2 pentads) would most likely need to be accompanied by very high temperatures and wind to have conditions that would lead to impacts). I believe you do go on to mention this later, but it should be mentioned now as well). This is a fundamental flaw with defining flash drought by duration. Perhaps duration is a piece of the flash drought definition when it comes to onset, but not likely for duration as a drought may linger on after a rapid onset/intensification has taken place. You do seem to imply/focus on onset in Section 2.2, but I’m getting mixed messages in your methods/findings up to this point. The key phrasing here in Line 134 is “events that last for only one pentad.”

Response: We appreciate the reviewer’s insightful comment. As shown in Fig. R9, we find that a number of events experience a rapid decline to below the 20th percentile in SM but recover quickly, which should be excluded because one or two pentads are very unlikely to result in any impacts. Indeed, even if the drought lasts for a very short period of time (1 or 2 pentads), it may also affect the growth of crops if accompanied by high temperature and wind. Thus, we have added this discussion in the revised version of the manuscript. In addition, although we focus on the onset phase of flash drought, we have also used the length of duration (≥ 3 pentads) after SM declining to the 20th percentile to choose those detected events that

have an adverse impact. The relevant statements in methods have been revised as follows.

“These events cannot be viewed as flash droughts because only one pentad is inadequate to diminish crop productivity and yield. Perhaps even if the drought lasts for a short period of time (1 or 2 pentads), it may also affect the growth of crops if accompanied by high temperature and wind, but this situation is not within the scope of our discussion. Therefore, the use of the definition focusing solely on the duration of flash droughts cannot guarantee that all events satisfy the key characteristics (drought severity and flash) of flash droughts.”

Fig. R9 Variation in SM percentiles for each grid point of flash droughts at 1-, 2-, 3-, 4-, and 5-pentad lead times without considering duration that can diminish crop productivity and yield (SM should not only drop below the 20th percentile but also last for at least three pentads).

Comment #9: Lines 191-92) Do you mean for “global” drought early warning indicators in regards to weekly/monthly updates? At the country level, a vast majority of indicators (although not for all countries) are available daily or weekly and at relatively high resolution (1-5km or better). The bigger issue is these triggers aren’t linked to adequate planning or decision making, or are simply ignored given the stigma of drought being a “slow-onset” development hazard. We know that simply isn’t true and this flash drought work is timely and relevant.

Response: According to the reviewer’s comment, we have revised the corresponding statements as follows.

“There are several products and indicators used to capture flash drought events, including the U.S. Drought Monitor (USDM), the Drought Impact Reporter (DIR), the U.S. Crops and Livestock in Drought, the Climate Prediction Center (CPC) SM, and the Vegetation Drought Response Index (VegDRI). Although some of them are available on a daily or weekly basis, but not for all countries. Thus, assessing the relatively fast onset development timescales of flash droughts can provide useful information for upgrading drought monitoring systems and indicators all over the world. To further improve the ability of monitoring and predicting flash droughts, the criterion of a rapid intensification rate should be taken into account in addition to the relatively short onset timescales for capturing the unique characteristics of flash droughts (rapid onset and rapid intensification).”

Comment #10: Lines 246-48) Again, I believe the tools are already there to develop the systems. Acceptance of rapid onset events and links to decision making and planning are the real gap. You are correct though in emphasizing the need to integrate these tools into our drought early warning systems. Sub-seasonal-to-seasonal (S2S) prediction needs in the context of drought are essential as well of course.

Response: We appreciate the reviewer’s positive comment. Given a flash drought’s onset timescale of only a few weeks, these are not sufficient for most monitoring and prediction products. Instead, products that are updated daily are required. This provides an opportunity to leverage synoptic weather forecasts in combination with seasonal forecasting efforts that have recently become available at shorter timescales, such as the SubX system (Pendergrass et al. 2020; Pegion et al. 2019).

References:

- Pendergrass, A. G. et al. Flash droughts present a new challenge for subseasonal-to-seasonal prediction. *Nat. Clim. Chang.* **10**, 191–199 (2020).
- Pegion, K. et al. The Subseasonal Experiment (SubX): a multi-model subseasonal prediction experiment. *Bull. Am. Meteorol. Soc.* **100** (10), 2043–2060 (2019).

Comment #11: Line 260) Do you mean 20th percentile and not 2th, or do you mean 2nd?

Response: We regret for the unclear statement, and have revised this statement as follows.

“More flash drought events can be identified when the threshold decreases to the 30th percentile, but such a decline (< the 5th percentile) in SM from the 30th to the 20th percentile may not be viewed as a rapid onset under the worst-case scenario (one month).”

Comment #12: Line 265) You should state clearly what your depth levels are, particularly the

“top” layers vs. the “root zone”. Most may have a general idea of what you mean by root zone (1m or so), but top could vary wildly depending on models and such. How many cm should be listed here and not just in a figure.

Response: According to the reviewer’s suggestion, the depths of root zone and top layer soil moisture have been listed as follows.

“GLEAM dataset:

The Global Land Evaporation Amsterdam Model (GLEAM) version 3.1a provides observationally constrained global daily surface (0–10 cm) and root-zone (10–250 cm) SM spanning the period from 2000 to 2020. The GLEAM surface SM was produced by employing an improved SM data assimilation system using three independent SM datasets, including two satellite-based SM products from the European Space Agency (ESA) Climate Change Initiative (CCI) and Soil Moisture Ocean Salinity (SMOS) and the surface SM from the Noah model in the Global Land Data Assimilation System (GLDAS). The root-zone SM was modelled from a three-layer water balance module with the input of precipitation infiltration and the outputs of evapotranspiration and drainage. The forcing data of the water-balance module includes Multi-Source Weighted Ensemble Precipitation (MSWEP) precipitation, ERA-Interim radiation and air temperature, CCI-LPRM vegetation optical depth, ESA CCI and GLDAS-Noah SM, etc.

GLDAS-2 dataset:

NASA GLDAS-2 (Global Land Data Assimilation System Version 2) models include VIC, Noah, and Catchment land surface models (CLSM) with a spatial resolution of 1.0 degree and a temporal resolution of three hours for 2000–2020. The goal of the GLDAS is to ingest satellite- and ground-based observational data products, using advanced land surface modeling and data assimilation techniques, in order to generate optimal fields of land surface states and fluxes. Daily surface (0–10 cm) and root-zone (0–200 cm) SM were obtained from VIC, Noah, and CLSM.”

Comment #13: Lines 297-98) Again, may want to tone down the “first time” language given Koster work.

Response: According to the reviewer’s comment, we have added the Koster’s work and have removed “first time” in this sentence.

“A previous study produced a global map of flash droughts, with a focus on the contribution of evapotranspiration (ET) and P to the occurrence of flash droughts⁴⁰. Nonetheless, few studies explore the rapid onset phase of flash droughts and underlying causes of the rapid onset which is the most important characteristic of flash droughts.”

Comment #14: Line 296) Seems a bit disjointed to read this and have the Conclusions come before both the Results and Discussion as well as the Methods? A lot of questions could be answered by putting the Methods earlier.

Response: According to the reviewer's constructive comment, we have adjusted the structure of this paper to some extent. However, the section of Methods is still placed after Results and Discussion according to the journal's format requirement.

Responses to Figures:

Comment #1: Figure 1: Too many graphics in this figure. The world map views don't project well and it is hard to see colors/patterns even in the defined red box regions on the maps. Same goes for the soil moisture plots all of these need to be larger and/or broken out into two separate figures at a minimum. Even at 125% magnification, they aren't very readable.

Response: According to the reviewer's comment, we have redrawn the figure to improve the readability.

Comment #2: Figure 3: Same issue as figure 1, but not as bad given only 3 global map views. Still, not very readable except for the pentad bar charts. Please make them larger.

Response: According to the reviewer's comment, we have redrawn the figure to improve the readability.

Comment #3: Figure 4f: needs to be larger.

Response: According to the reviewer's suggestion, we have made all figures larger.

Responses to Supplementary Material:

Comment: Figure 1 a-j: Not readable. Too small. Separate Figure 1 needed only showing a few world maps at a time so we can see the patterns better. Figure 10 is much better

Figure 8: Although only showing three global maps, they are still too small and hard to read even at 125% magnification. Need to make them bigger and perhaps only stack on top of each other instead of side-by-side? Figure 10 is readable as an example.

Figure 9: stack and make larger please...like Figure 10

Response: According to the reviewer's suggestion, we have redrawn all figures to improve the readability.

Responses to Reviewer #3:

General comments:

The premise of this study is interesting and, in my opinion, is worth exploring and potentially impactful. The question if flash droughts are intensifying more rapidly is of great interest for improving drought monitoring/early warning and reducing community vulnerability. However, there are numerous shortcomings and limitations in the present study that leave me wondering how robust and relevant the results and conclusions are for the stated purpose. Below is a list of my concerns, the largest of which is that the results of increase 3-pentad flash drought frequency are not linked to any physical change in regional climate, but are instead explained by vague references to changes in temperature and aridity. My recommendation is that the authors significantly revise the current study to focus on (1) quantifying the robustness of the observed trends and (2) analyzing the causes of the observed increase in 3-pentad flash drought events globally.

Response: We appreciate the reviewer's constructive comments and suggestions. According to the reviewer's comment, we have significantly revised the current study with a focus on quantifying the robustness of the observed trends and analyzing the causes of the rapid onset of flash droughts. Our responses to the reviewer's specific comments are shown as follows.

Specific comments:

Comment #1: Lines 35-38) At least for the references to the USA, neither of these studies pointed to climate change as a primary forcing of the 2012 or 2017 flash droughts, nor propose that flash droughts are increasing in frequency because of global warming. In fact, Hoerling et al. (2014) specifically list decadal variability as an important driver of the risk of 2012-like drought in the central U.S.

Response: We regret for the inaccurate statement, and have revised the corresponding statement in the revised version of the manuscript.

“For example, the 2012 flash drought that occurred over the Central United States caused a tremendous impact on agricultural production and the economy, with \$12 billion losses attributed to this event⁵. Furthermore, analysis of retrospective climate simulations found virtually no dry signal, owing to the unusual speed and intensity (developed in May and reached peak intensity by August), over the major corn-producing regions of the eastern Great Plains where this drought event resulted in major curtailment of corn crop yields. The flash drought event that occurred over Southern Queensland, Australia in 2018 de-vegetated the landscape and drove livestock numbers to the lowest level in the country⁶. And some countries experienced more frequent flash droughts in recent years, including southern China^{7,8,9}, USA^{10,11}, and Africa¹².”

Comment #2: Lines 164-165) According to the U.S. Department of Agriculture, Alaska's agricultural product in 2006 (last survey) resulted in over \$60 million in cash receipts: https://www.nass.usda.gov/Statistics_by_State/Alaska/About_Us/index.php. So, there is agriculture there and certainly in parts of the Sahara. Because this paper is not solely focused on flash drought impacts to agriculture, why exclude these regions because they are not

agriculturally productive relative to other regions? Certainly a flash drought could have impacts on water resources or permafrost development in Alaska.

Response: According to the reviewer's comment, we have included Sahara and Alaska for all assessments in the revised version of the manuscript.

Comment #3: Lines 192-195) It looks from Figure 2 that the majority of flash droughts in each region occur in 3 pentads, not 1 or 2. But by grouping 1, 2, and 3 pentad developments together as you do here, it implies these events occur much more quickly. This is restated in lines 304-306. As most drought monitoring products are updated daily or weekly, I think they are capable of monitoring and maybe providing early warning for events that develop over a 15 day period.

Response: We appreciate the reviewer's constructive comment. To better assess the onset development timescales of flash droughts, we have revised the pentad-based groups. Specifically, all flash droughts have been classified into three different types, including flash droughts developing in 1 pentad, 2-3 pentads, and 4-5 pentads. And we find that the proportion of 1-pentad flash droughts in all flash drought events shows a significant ($P < 0.01$) increasing trend globally during the whole period 2000–2020 for all three datasets (Fig. R10). The magnitude of estimated slope suggests an increase of 23.46–29.09% in the proportion of 1-pentad flash droughts during the study period for all three datasets, with an annual growth of 1.12–1.39% (Fig. R10c). This implies more flash drought events developing in 1 pentad (5 days). There are certain products and indicators available on a daily or weekly basis, but not for all countries. Thus, assessing the relatively fast onset development timescales of flash droughts can provide useful information for upgrading drought monitoring systems and indicators all over the world. To better address the reviewer's comment, we have revised the relevant statements as follows.

“We used the nonparametric Mann-Kendall statistic to examine changes in the percentages of flash droughts at different lead times relative to all flash drought events at an annual timescale. Both the trend and the number of flash drought events developing in 2 and 3 pentads, derived from all three models, are consistent and match well with each other. In addition, the number of flash droughts developing in 5 pentads is too small to perform statistical significance tests over a 21-year period, and thus we incorporated the events at 2- and 3-pentad lead times into one category (flash droughts developing in 2-3 pentads) and the events at 4- and 5-pentad lead times into another category (flash droughts developing in 4-5 pentads). We find that the proportion of 1-pentad flash droughts in all flash drought events shows a significant ($P < 0.01$) increasing trend globally during the whole period 2000–2020 for all three datasets. The magnitude of estimated slope suggests an increase of 23.46–29.09% in the proportion of 1-pentad flash droughts during the study period for all three datasets, with an annual growth of 1.12–1.39% (Fig. 2c). Over most regions, the 1-pentad flash droughts also show an increasing trend, especially in Southern South America, Southeast Asia, Central North America, Southern Africa, and Eastern Africa where the increase in the proportion of 1-pentad flash droughts is up to 20% (Supplementary Fig. 5), indicating that more flash drought events develop within one week over these regions. By contrast, the proportions of flash droughts developing both in 2-3 and 4-5 pentads, derived from all three datasets, show a statistically significant ($P <$

0.05) decrease from 61.93–68.06% and 16.14–27.83% in 2000 to 53.05–57.51% and 4.89–10.84% in 2020, implying that flash droughts take less time to develop (Fig. 2d, e). Similar trends can also be found in most sub-regions, with the largest decreasing trend over South Asia, followed by South Africa, Amazon Basin, Southeast Asia, and Central North America (Supplementary Figs. 6 and 7). More importantly, we find that there is no consistently significant increasing trend in the frequency of all flash droughts, with only one dataset showing a significant increasing trend (Supplementary Fig. 8). And over most regions the frequency of flash droughts, in general, does not show a statistically significant trend, and there is only a significant increasing trend over the Mediterranean Basin, Central Asia, and Sahara (Supplementary Fig. 9).”

Fig. R10 Comparison of percentages of flash droughts at different lead times relative to all flash droughts and their temporal evolutions. a The global proportion of flash droughts at different lead times relative to all flash droughts. The red dashed lines at the top of each bar represent the range of uncertainty in three different datasets. **b** The proportion of flash droughts at different lead times relative to all flash droughts over 21 regions. **c**, **d**, and **e** Temporal evolutions of percentages of flash droughts with the lead times of 1-, 2-3, and 4-5 pentads relative to all flash droughts.

Supplementary Fig. 5 Temporal evolution of the percentage of flash droughts developing at 1 pentad relative to all flash droughts across different regions of the world. The red line represents the mean of results obtained from three datasets, and the grey shadows represent the ranges of results derived from three datasets.

Supplementary Fig. 6 Temporal evolution of the percentage of flash droughts developing at 2-3 pentads relative to all flash droughts across different regions of the world. The red line represents the mean of results obtained from three datasets, and the grey shadows represent the ranges of results derived from three datasets.

Supplementary Fig. 7 Temporal evolution of the percentage of flash droughts developing at 4-5 pentads relative to all flash droughts across different regions of the world. The red line represents the mean of results obtained from three datasets, and the grey shadows represent the ranges of results derived from three datasets.

Supplementary Fig. 8 Temporal evolution of the frequency of flash droughts all over the world. The red line represents the mean of results obtained from three datasets, and the grey shadows represent the ranges of results derived from three datasets.

Comment #4: Line 298) this really isn't a "new" definition, but mostly an adoption of a definition that's been around for 5-6 years now. Give credit to the definitions you adopted here.

Response: According to the reviewer's comment, we have changed the "new" definition to the "intensification rate-based" definition, and the "old" definition to the "duration-based" definition in the revised version of the manuscript.

Comment #5: Line 298) The finding of increased frequency of 3-pentad development in place of 4- or 5-pentad development is interesting and noteworthy; however, the current study provides no satisfying explanations as to why this is happening at a global scale. Lines 132-160 (which I think could be removed to the benefit of the study) only describe a typical monsoonal climate and the atmospheric driver of essentially all warm season droughts in the central U.S. This discussion does not discriminate the factors at play specifically for flash drought in these regions, nor does it suppose why an increase in more rapid onset droughts has occurred over the last 20 years. Lines 221-232 provide some description of the meteorological conditions associated with flash drought, but again do not go far enough to explaining the reported increasing trends. This also brings into account the issue of no tests for statistical significance. The shaded time series plots in Figure 4 show quite a bit of variability. Could you at least report the calculated p-values to give the reader an idea of the confidence in the trend relative to the time series noise? This is particularly important given the relatively short study period.

Response: We appreciate the reviewer's constructive comment. According to the updated analysis, our findings reveal that the proportion of 1-pentad flash droughts in all flash drought

events shows a significant ($P < 0.01$) increasing trend globally during the whole period 2000–2020 for all three datasets. Thus, we have performed an in-depth analysis of why flash droughts tend to develop with a shorter timescale. Specifically, we have compared the anomalies of annual flash drought-affected land areas, annual mean T, annual total P, annual mean VPD, and relative SM change over the NEU-MEU (the total area of Northern Europe and Mediterranean Basin) during 2000–2020 to identify potential factors leading to the rapid onset of flash droughts. To strengthen the robustness of main findings, we have added a new data source (GLEAM) and the uncertainty bounds estimated based on three different datasets. In addition, we have calculated P-values of all trends to give the reader an idea of the confidence relative to the time series noise (see Fig. R10). The following discussions related to attribution analysis have been added to the revised version of the manuscript.

“To explore flash drought evolution characteristics and to conduct attribution analysis, we plotted the time series of anomalies of annual flash drought-affected land areas, annual mean T, annual total P, annual mean VPD, and relative SM change over the NEU-MEU (the total area of Northern Europe and Mediterranean Basin) during 2000–2020 (Fig. 3). These two places cover most of the European region, and the annual frequencies of flash droughts of Northern Europe and Mediterranean Basin show an increasing trend (Supplementary Fig. 9). It is found that high T, low P, and high VPD often coincide with a high percentage of flash drought-affected land areas. The high T, low P, and high VPD in 2007, 2014, 2016, and 2018 rank in the top 5 during 2000–2020 (Fig. 3a-d), and the percentage of land areas in 2007, 2014, 2016, and 2018 also rank in the top 5 (red boxes in Fig. 3). On the other hand, there are few significant correlations between T and the percentage of flash drought-affected land areas. The high percentage of flash drought areas is not always accompanied by high T, but it is always under the P-deficit and VPD condition (black box in Fig. 3). This suggests that the flash drought occurrence is not dominated by T. The high T can cause SM to drop, but such a decrease in SM anomalies is not large¹⁴. The low P and high VPD before or during the onset of flash drought events are needed to increase the SM deficits and to make the conditions more favorable for the occurrence of flash droughts. In this sense, P and VPD play an important role in enhancing the possibility of flash drought occurrence. In addition, the variation of annual mean VPD is similar to the variation of relative change in annual mean SM. Specifically, the relative change in SM increases (decreases) when VPD increases (decreases), suggesting that the increasing VPD triggers a rapid decline in SM. And the rapid decline in SM induces the occurrence of a flash drought event. Thus, high T, low P, and high VPD create a flash drought-prone environment, and VPD is a significant factor influencing the speed of onset, which tends to drive a rapid decline in SM to cause flash drought events.”

Fig. 3 Interannual variations of flash drought-affected land area and its related variables over the NEU-MEU from 2000 to 2020. a, e Anomalies of flash drought-affected land area and relative SM change over the NEU-MEU from three different datasets. The relative SM change is defined as the change of SM percentiles between adjacent pentads (the SM percentile at the current pentad minus the SM percentile at the next pentad), where positive values represent a decline in SM at the current pentad. **b, c, and d** Anomalies of annual mean T, annual total P, and annual mean VPD over the NEU-MEU.

Comment #6) Somewhat related to the last comment, without any notion of the physical causes of reported changes in flash drought intensification rate, it is difficult to assess where and how to improve operational drought monitoring and prediction. The statement on lines 310-313 and

similar ones throughout the paper are quite vague and not really helpful to improving flash drought early warning. The majority of operational drought monitoring products are updated daily or weekly and can therefore capture a 3-, 4-, or 5-pentad flash drought. Can the authors demonstrate what is causing the reported increase in more rapid flash drought intensification and give some more relevant information for improving drought monitoring and prediction?

Response: We appreciate the reviewer’s insightful comment. In addition to the attribution analysis from a regional perspective, as mentioned in our previous response, we have also detected the cause of rapid onset of flash droughts through land–atmosphere interactions from a global perspective. We find that the relative change in SM increases (decreases) when VPD increases (decreases), suggesting that the increasing VPD triggers a rapid decline in SM. And the rapid decline in SM induces the occurrence of a flash drought event. In other words, VPD is a significant factor influencing the speed of onset, which tends to drive a rapid decline in SM to cause flash drought events. Thus, we have further investigated the relationship between VPD and flash droughts. As shown in Fig. R11, we find that the strong coupling of SM and VPD through land-atmosphere feedbacks contributes to the rapid decline in SM. Moreover, the positive correlation between the frequency of flash droughts occurring in 21 regions and PMF will increase to 0.75 when the threshold of extreme SM decreases to the 20th percentile, suggesting that the intensity of SM-VPD coupling will increase when there is a further decline in SM. The following discussions have been added to the revised version of the manuscript.

“There is a strong negative correlation between SM and VPD, and the low SM is always accompanied by the high VPD⁶⁰. In turn, the high VPD enhances the atmospheric evaporative demand, which can exacerbate the decline in SM through land–atmosphere interactions⁶¹. To investigate the contribution of VPD to the occurrence of flash droughts, we examined the underlying relationship between SM and VPD (Fig. 4). We find that SM is strongly correlated with VPD, which is evident in the bimodal distribution of SM and VPD: high VPD and low SM, and low VPD and high SM (Fig. 4b). To further explore the onset phase of flash droughts, we defined the bivariate extremes as those with SM below the 30th percentile (the 30th percentile was used to represent the mean condition of flash drought onset phase) and VPD above the 90th percentile. Probability multiplication factor (PMF) was calculated to assess the increasing frequency of compound events compared to those expected when SM and VPD were independent, and a PMF of 1 represents no increase in the joint probability. We find that certain regions with a high value of PMF are consistent with those experiencing a high frequency of flash drought events (dashed boxes in Fig. 1c and Fig. 4a). And even if the thresholds of SM and VPD change, similar patterns can also be obtained (Supplementary Fig. 10). We also calculated the PMF of SM-T and SM-P, and the corresponding patterns show less correlation with those of flash drought frequency compared with SM-VPD (Supplementary Fig. 11). More importantly, a positive correlation ($R=0.64$, the point of Tibet was excluded from correlation analysis) between the frequency of flash droughts occurring in 21 regions and PMF is found (Fig. 4c), especially for the Southeast Asia, Amazon Basin, and Southern South America, and Eastern North America where the joint probability of bivariate extremes is approximately 1.84–2.55 times higher than the probability of the independent combination. This indicates that the strong coupling of SM and VPD through land-atmosphere feedbacks contributes to the rapid decline in SM. Moreover, the positive correlation between the frequency of

flash droughts occurring in 21 regions and PMF will increase to 0.75 when the threshold of extreme SM decreases to the 20th percentile, suggesting that the intensity of SM-VPD coupling will increase when there is a further decline in SM (Fig. 4d).”

Fig. R11 Relationship between SM and VPD. **a** Mean PMF of concurrent extreme SM (below the 30th percentile) and VPD (above the 90th percentile) for the period 2000–2020 based on three datasets. **b** Mean probability of percentile bins of SM and VPD across all land grid points. **c** Spatial relationship between the frequency of occurrence of flash droughts and the PMF of SM and VPD extremes (below the 30th percentile SM and above the 90th percentile VPD). **d** Spatial relationship between the frequency of occurrence of flash droughts and the PMF of SM and VPD extremes (below the 20th percentile SM and above the 90th percentile VPD). Each point in **c** and **d** is the mean of the results from three datasets.

Comment #7: Lines 12-13) Despite the argument that flash drought causes more impacts being used is numerous studies, it has actually never been explicitly studies or demonstrated (at least to my knowledge). Inarguably, flash drought provides less warning time for preparation, but I have not seen an objective evaluation of disproportionate impacts from the lack of preparation relative to a slower evolving drought. I don't think this study would be weakened by removing this statement.

Response: According to the reviewer’s comment, we have revised the corresponding statement as follows.

“The emergence of flash drought has attracted widespread attention due to its rapid onset and devastating impacts on the environment and agriculture. However,

little is known about the recent evolution of flash droughts in terms of the speed of onset and the causes of such a rapid onset phase of flash droughts.”

Technical edits:

Comment #1: Line 24) presumably, flash droughts have occurred prior to their scientific recognition. I suggest rephrasing to "a newly defined drought type..."

Response: According to the reviewer’s suggestion, we have revised the corresponding statement as follows.

“Flash droughts, as a newly defined drought type, have been receiving increasing attention from the scientific community in recent years.”

Comment #2: Line 29) "no early warning" should really be "less early warning". Also line 32: there was some early warning in 2012, just not as much as other droughts.

Response: According to the reviewer’s comment, we have revised the corresponding statement as follows.

“Since flash droughts can be distinguished by their rapid development, there is less early warning for impact preparation, potentially causing more severe impacts on agriculture and society than the slowly-evolving droughts^{3,4}.”

Comment #3: Line 49) "access" should be "assess"

Response: We have made a correction as follows.

“These two types of definitions of flash droughts provide us with alternative approaches to assess this extreme phenomenon from different perspectives (duration and intensification rate).”

Comment #4: Figure 1) why are the methods referred to as "new" and "old"? They were developed within 3-4 years of each other.

Response: To address the reviewer’s comment, we have changed the “new” definition to the “intensification rate-based” definition, and the “old” definition to the “duration-based” definition in the revised version of the manuscript.

Comment #5: Line 135) "mostly" should be "most"

Response: We have made a correction as follows.

“Nonetheless, both methods (definitions) indicate that flash droughts are most likely to occur in humid and semi-humid regions (e.g., Southeast Asia, East Asia, Amazon Basin, Eastern North America, and Southern.”

Comment #6: Line 174) Do you mean that the intensification rate was 28.6 percentiles?

Because the 28.6th percentile refers to a specific percentile of a distribution, not how many percentiles a condition has increased or decreased.

Response: We regret for the confusing statement. It does mean that the intensification rate was 28.6 percentiles, and thus we have revised the corresponding statement in the revised version of the manuscript.

Comment #7: Lines 178-180) there is some circular logic here. You defined flash droughts as those in which onset development occurs in 5 pentads or fewer, then find that these events onset more quickly than traditional droughts. Of course they do, that's how they were defined here.

Response: We appreciate the reviewer's comment. In this paper, we explored the rapid onset of flash droughts by analyzing the timescale of flash drought onset phase and revealing how fast flash droughts develop over the past 21 years from 2000 to 2020. According to the reviewer's comment, we realize that there is actually no need to mention the traditional droughts. Of course, the comparison between traditional droughts and flash droughts is a meaningful topic, but it is not the focus of this paper. Thus, we have removed the relevant statement related to traditional droughts in the revised version of the manuscript.

Comment #8) What is the trend unit here? Is it flash drought events per year, per decade?

Response: We regret for the unclear statement. The unit of trend is "flash drought events per year", and we have added the corresponding description in the caption of Fig. 2.

Comment #9: Line 331) can you provide more detail (perhaps in supplemental material) about the percentile calculation? Did you remove any seasonal cycle of soil moisture prior to calculating percentiles? This will be important in regions with pronounced seasonal drying and wetting.

Response: We regret for the unclear statement. In this paper, percentiles were determined for every grid independently throughout the same pentad for each year over the period of 2000 to 2020, to enable a comparison of relative SM change throughout the same time every year. Since percentiles calculated based on the same pentad for years can avoid the influence of seasonal changes, there is no need for deseasonalization of estimated SM. According to the reviewer's suggestion, we have added more details about the percentile calculation in supplemental material.

Comment #10: Lines 359-364) the duration component is important to include, I'm glad that was done here. However, did you also include a duration component for flash drought recovery? For example, I can imagine a situation where the soil moisture declines to below the 20th percentile, remains there for 2-3 pentads, recovers quickly to the 40th percentile, and remains there for 1 pentad before quickly returning to the 20th or below. In that situation is there a single flash drought, or 2 flash droughts?

Response: We appreciate the reviewer's constructive comment. We have considered the situation where the soil moisture declines to below the 20th percentile and recovers quickly to the 40th percentile, and then remains there for 1 pentad before quickly returning to the 20th or

below. By our definition, the event in this case can be viewed as a drought event, but not a flash drought because such an event does not experience a rapid onset phase that SM should decrease from the 40th percentile to the 20th percentile within a month.

Reviewers' Comments:

Reviewer #1:

Remarks to the Author:

The Authors have addressed my concerns. This MS can be published in the present form.

LS

Reviewer #2:

Remarks to the Author:

I'd like to thank the authors for their taking the time to provide such thorough responses (with associated follow-up work) in responding to all of the reviewer comments. I feel "most" of my initial first-round review comments were adequately addressed and explained by the authors. I do think overall that the figures have been improved and are more readable.

However, a couple of things still stand out as needing to be addressed. The first being the abstract and reference to the 2012 drought costing only \$12B (Hoerling et al.) Instead I would recommend using the NCEI Billion Dollar Disaster number of ~\$35.7B USD, which is widely accepted to be the standard catalog of disaster estimates for all hazards. <https://www.ncdc.noaa.gov/billions/>

As mentioned in the first round of review, the statement that this is the "first". The team did reference the Koster et al. work, but still choose to use first when it comes to global studies when it really should say "one of the first..."

Finally, from a validation standpoint, it is hard to grasp there being a "drought" (say vs. a rapid soil moisture depletion) after one, two or even three pentads. If there aren't impacts to the environment/crops then is there a drought? Natural buffers and resilience could see a rapid drop in soil moisture levels, but still see a "lag" in the response prior to impacts manifesting themselves or maybe no impacts will occur at all. Drought is more complicated than this simplified example.

Whether this paper is published in Nature COMMS or not, it is now worthy of publishing somewhere given the extensive work and changes the author team took in addressing the reviewers feedback and comments. In my opinion, it can still contribute to the field in this fast evolving area of flash drought.

Reviewer #3:

Remarks to the Author:

I appreciate the additional work and effort the authors have put into this study, responding to all reviewers' comments. The analysis of VPD and soil moisture coupling in the context of flash drought, in particular, is a great addition. With that said, my lone remaining concern is that the additional work did not provide a physical basis for the apparent increasing trend in 1-pentad flash droughts, which is a primary finding of the study. I think the impact of this study is dependent on linking this trend to a physical change in regional climate - perhaps increasing T, VPD, etc.? To be clear, the only addition I'd like to see prior to acceptance (other than a few minor technical corrections specified below) is attribution of the 1-pentad flash drought trends in most global regions to a change in those regions' climates.

Technical corrections:

1) Abstract, line 23: you should specify what has an "increasing trend globally".

2) Abstract: one important finding that is omitted from the abstract is that flash droughts don't appear to be occurring more frequently in most global regions, just coming on faster

3) Line 27: I think you should delete "as a new drought type", because it is a term that has been around for nearly 20 years now

4) Lines 45-46: edit to "flash droughts as proposed by Mo..."

5) Line 49: add "definition" after "Another"

6) Line 106: delete "that have a large impact on the environment". The abnormally dry class is mainly used to delineate areas that are going into or coming out of drought. If there are significant environmental impacts, the region is typically classified at a more severe level (i.e., D1, D2)

7) Line 154: I think the term "lead time" is imprecise. You're really looking at the intensification time of the flash droughts, whereas lead time implies some sort of forecasting component.

8) Line 176-177: I understand the label of 1-pentad flash droughts, but unfortunately it implies a flash drought that lasts 1 pentad (i.e., a duration of 1 pentad). I don't have a good solution for this issue, maybe labeling "1-pentad onset flash droughts"? This may seem pedantic, but there has been quite a lot of poor science and miscommunication around flash drought in recent years, so it's important to be as clear and precise as possible.

9) Line 212: replace "region" with "continent"

Black font = Reviewers' comments
Blue font = Response to reviewers' comments
Red font = Changes in the text of the manuscript

Responses to Reviewer #1:

General comments:

The Authors have addressed my concerns. This MS can be published in the present form.

Response: We appreciate the reviewer's positive comment.

Responses to Reviewer #2:

General comments:

I'd like to thank the authors for their taking the time to provide such thorough responses (with associated follow-up work) in responding to all of the reviewer comments. I feel "most" of my initial first-round review comments were adequately addressed and explained by the authors. I do think overall that the figures have been improved and are more readable.

Response: We appreciate the reviewer's positive comment.

Specific comments:

Comment #1) However, a couple of things still stand out as needing to be addressed. The first being the abstract and reference to the 2012 drought costing only \$12B (Hoerling et al.) Instead I would recommend using the NCEI Billion Dollar Disaster number of ~\$35.7B USD, which is widely accepted to be the standard catalog of disaster estimates for all hazards. <https://www.ncdc.noaa.gov/billions/>

Response: We appreciate the reviewer's careful review, and we have used the NCEI Billion Dollar Disaster number of ~\$35.7B USD in the revised version of the manuscript.

"For example, the 2012 flash drought that occurred over the Central United States caused a tremendous impact on agricultural production and the economy, with about \$35.7 billion losses attributed to this event⁵."

Reference:

NOAA National Centers for Environmental Information (NCEI) US Billion-Dollar Weather and Climate Disasters. 2017. Available online: <https://www.ncdc.noaa.gov/billions/> (accessed on 1 June 2021).

Comment #2) As mentioned in the first round of review, the statement that this is the "first". The team did reference the Koster et al. work, but still choose to use first when it comes to global studies when it really should say "one of the first..."

Response: We regret for the overstatement of "first", and we have removed all the related

statements in the revised version of the manuscript.

“Here, we present a comprehensive assessment of the onset development timescales of flash droughts and the underlying mechanisms on a global scale.”

“This global study is to shed light on the onset development timescales and the causes of the rapid onset speed, advancing our understanding of the evolution tendency of flash droughts and providing insights into the implementation of flash drought forecasts and early warning systems.”

Comment #3) Finally, from a validation standpoint, it is hard to grasp there being a “drought” (say vs. a rapid soil moisture depletion) after one, two or even three pentads. If there aren't impacts to the environment/crops then is there a drought? Natural buffers and resilience could see a rapid drop in soil moisture levels, but still see a “lag” in the response prior to impacts manifesting themselves or maybe no impacts will occur at all. Drought is more complicated than this simplified example.

Response: We appreciate the reviewer’s constructive comment. We agree with the reviewer that the captured flash drought events should have impacts on the environment/crops and being a "drought" after one, two or even three pentads may not result in severe impacts. To adequately reflect flash drought intensity and duration that can diminish crop productivity and yield, in our definition, SM should not only drop below the 20th percentile but also last for at least three pentads ($t_p - t_e \geq 3$). Thus, the duration of flash droughts captured in our study is at least 5 pentads (25 days). This criterion can be used to exclude those events that decrease from above the 40th percentile to below the 20th percentile and then recover quickly up to the 40th percentile probably due to the natural buffers and resilience. As shown in Fig. R1, a number of events experienced such a rapid decline in SM but recovered quickly, which were excluded in our study.

Fig. R1 Variation in SM percentiles for each grid point of flash droughts at 1-, 2-, 3-, 4-, and 5-pentad onset times without considering the duration that can diminish crop productivity and yield (SM should not only drop below the 20th percentile but also last for at least three pentads). The red boxes highlight the events with SM decreasing to below the 40th percentile occur, but then rapidly recover up to the 40th percentile within only one or two pentads under abnormally dry conditions.

Part of the impact of drought comes from its longer duration, which is the same for flash drought. Thus, we evaluated the duration of flash droughts captured based on three datasets. We find that the mean duration of flash droughts is 16.01, 13.82, and 12.30 pentads for Noah, GLEAM, and CLSM models, respectively. About 65.82% (Noah), 64.88% (GLEAM), and 60.01% (CLSM) of flash droughts last for longer than 1 month (6 pentads), as shown in Fig. R2. According to the study by Otkin et al. (2018), the 2012 flash drought adjacent to the Marena Mesonet station caused the grasses to rapidly turn brown and go dormant over a 6-week period, which stands in sharp contrast to the continued greenness over the same period in 2014. And the flash drought event that occurred across the northern high plains in 2017 increased in drought severity over a 2-month period and sharply reduced wheat yields across the region (Otkin et al., 2018). Thus, the duration of flash droughts captured in our study may be able to guarantee that most flash droughts can have impacts on the environment/crops from the perspective of flash drought duration.

Reference:

Otkin, J. A. et al. Flash droughts: a review and assessment of the challenges imposed by rapid-onset droughts in the United States. *Bull. Am. Meteor. Soc.* **99**, 911–919 (2018).

Fig. R2 Spatial pattern of flash drought duration based on different models. a–d Mean duration of occurrence of flash droughts based on Noah, CLSM, GLEAM, and the ensemble of results from three models. **e** Probability density of the mean duration of flash droughts based on Noah, CLSM, GLEAM, and the ensemble of results from three models.

On the other hand, the impact comes not only from the duration of flash droughts but also from its rapid onset phase. Previous studies demonstrated that suddenness with environmental anomalies during the onset phase of flash drought could have impacts on the economy and local ecosystems (Svoboda et al., 2002; Otkin et al., 2018; Pendergrass et al., 2020). Compared with before (two pentads prior to flash drought) and recovery phases of flash drought, the onset phase is often accompanied by varying meteorological conditions. Fig. R3 shows the meteorological conditions during before, onset and recovery phases of flash drought, as well as the difference between different phases of flash drought. We find that when it varies from the before phase to the onset phase, temperature (T) and vapor pressure deficit (VPD) show a considerable increase in most regions of the world and the change in VPD is more obvious than T, whereas the amount of precipitation (P) decreases. After the rapid onset of flash drought, nevertheless, T and VPD all drop a bit and the decreasing rate of P slows down during the

recovery phase. Similar results can also be obtained using different datasets, as shown in Figs. R4-6. This indicates that the onset phase of flash drought is likely to be accompanied by an obvious increase in T, VPD, and a decrease in P, leading to higher T, higher VPD, and lower P when compared with other phases of flash drought. Such sudden changes in T, VPD, and P during the onset phase of flash drought will put forward higher requirements for plants and humans adapting to the fast-changing environments compared with slowly evolving traditional droughts. In addition, the simultaneous occurrence of high T, high VPD, and P deficit may be the primary reason for the impacts caused by the rapid onset of flash drought. For example, the tendency for high VPD and low SM events to co-occur may cause drought- and heat-driven reductions in vegetation productivity to be much greater than if VPD and SM do not covary (Zhou et al., 2019). The simultaneous occurrence of extreme high T, high VPD, and P deficit may contribute to certain hazards that compound the impacts of flash drought, such as heatwaves, wildfires and soil erosion. These may induce public-health effects of heat stress or air-quality degradation due to forest fires (Pendergrass et al., 2020). Even after the physical cause of a flash drought (e.g., anomalous meteorological conditions) has been eliminated in a region, an area may still experience lingering hydrological impacts for months or years, depending on the timing, duration, and intensity of flash droughts (Svoboda et al., 2002).

Overall, in addition to the impacts of flash drought caused by a longer duration, the impacts of flash drought are not only aggravated by a rapid change in meteorological conditions, such as an obvious increase in T and VPD and a decrease in P, but also by the compound effects of climatic factors, such as the compound effect of high VPD and low SM.

References:

Svoboda, M. et al. The Drought Monitor. *Bull. Am. Meteorol. Soc.* **83**, 1181–1190 (2002).

Otkin, J. A. et al. Flash droughts: a review and assessment of the challenges imposed by rapid-onset droughts in the United States. *Bull. Am. Meteor. Soc.* **99**, 911–919 (2018).

Pendergrass, A. G. et al. Flash droughts present a new challenge for subseasonal-to-seasonal prediction. *Nat. Clim. Change* **10**, 191–199 (2020).

Zhou, S. et al. Projected increases in intensity, frequency, and terrestrial carbon costs of compound drought and aridity events. *Sci. Adv.* **5**, 1, eaau5740 (2019).

Fig. R3 Meteorological conditions during before (two pentads prior to the occurrence of flash drought), onset and recovery phases of flash drought, as well as the difference between different phases of flash drought in light of flash droughts captured based on SM from Noah, CLSM, and GLEAM models. a–c T, VPD, and P for the before phase of flash drought. d–f Same as a–c but for the onset phase of flash drought. g–i Same as a–c but for the recovery phase of flash drought. j–l Differences of T, VPD, and P between before and onset phases of flash drought. m–o Differences of T, VPD, and P between onset and recovery phases of flash drought. These results are the ensemble of the results from three datasets.

Fig. R4 Meteorological conditions during before (two pentads prior to the occurrence of flash drought), onset and recovery phases of flash drought, as well as the difference between different phases of flash drought in light of flash droughts captured based on SM from Noah model. a–c T, VPD, and P for the before phase of flash drought. d–f Same as a–c but for the onset phase of flash drought. g–i Same as a–c but for the recovery phase of flash drought. j–l Differences of T, VPD, and P between before and onset phases of flash drought. m–o Differences of T, VPD, and P between onset and recovery phases of flash drought.

Fig. R5 Meteorological conditions during before (two pentads prior to the occurrence of flash drought), onset and recovery phases of flash drought, as well as the difference between different phases of flash drought in light of flash droughts captured based on SM from CLSM model. a–c T, VPD, and P for the before phase of flash drought. d–f Same as a–c but for the onset phase of flash drought. g–i Same as a–c but for the recovery phase of flash drought. j–l Differences of T, VPD, and P between before and onset phases of flash drought. m–o Differences of T, VPD, and P between onset and recovery phases of flash drought.

Fig. R6 Meteorological conditions during before (two pentads prior to the occurrence of flash drought), onset and recovery phases of flash drought, as well as the difference between different phases of flash drought in light of flash droughts captured based on SM from GLEAM model. a–c T, VPD, and P for the before phase of flash drought. d–f Same as a–c but for the onset phase of flash drought. g–i Same as a–c but for the recovery phase of flash drought. j–l Differences of T, VPD, and P between before and onset phases of flash drought. m–o Differences of T, VPD, and P between onset and recovery phases of flash drought.

Responses to Reviewer #3:

General comments:

I appreciate the additional work and effort the authors have put into this study, responding to all reviewers' comments. The analysis of VPD and soil moisture coupling in the context of flash drought, in particular, is a great addition. With that said, my lone remaining concern is that the additional work did not provide a physical basis for the apparent increasing trend in 1-pentad flash droughts, which is a primary finding of the study. I think the impact of this study is dependent on linking this trend to a physical change in regional climate - perhaps increasing T, VPD, etc.? To be clear, the only addition I'd like to see prior to acceptance (other than a few minor technical corrections specified below) is attribution of the 1-pentad flash drought trends in most global regions to a change in those regions' climates.

Response: We appreciate the reviewer's positive comment for the additional work, and we regret for the lack of the explanation of the physical basis for the apparent increasing trend in 1-pentad flash droughts. To address the reviewer's comment, we first compared meteorological conditions between different phases of flash drought including before (two pentads prior to flash drought), onset, and recovery phases (Fig. R7). We find that when it changes from the before phase to the onset phase, T and VPD show an obvious increase over most regions of the world and the change in VPD is more obvious than T, whereas the amount of P decreases. After the rapid onset of flash drought, however, both T and VPD drop a bit and the decreasing rate of P slows down during the recovery stage. This indicates that the onset stage of flash drought is likely to be accompanied by an obvious increase in T, VPD, and a decrease in P, leading to higher T, higher VPD, and lower P when compared with other stages of flash droughts. This is consistent with the main findings of our study that atmospheric aridity (higher T, higher VPD, and P deficit) is likely to create a flash drought-prone environment, and the changes in these climatic factors may be the main reason for the apparent increasing trend in 1-pentad flash droughts. Thus, we further compared the time series of the percentages of flash droughts with the 1-pentad onset time relative to all flash droughts, annual mean VPD, annual mean T, and annual total P for Northern Europe (NEU) and Mediterranean Basin (MED) because these two adjacent regions show different trends in the percentages of flash droughts with the 1-pentad onset time relative to all flash droughts. We find that the percentages of 1-pentad onset flash droughts show a significant ($P < 0.05$) increasing trend over NEU but an insignificant ($P > 0.05$) increasing trend over MED for three datasets (Fig. R8). According to the changes in the annual mean VPD, annual mean T, and annual total P over NEU and MED, we find that the significant increasing trend in the percentages of 1-pentad onset flash droughts over NEU is accompanied by a significant increasing trend in annual mean VPD and annual mean T, and a decrease in P, whereas MED experiences an insignificant ($P > 0.05$) increase in annual mean VPD (Fig. R9), suggesting that the significant increasing VPD may be the main contributor to the increasing trend of the percentages of flash droughts with the 1-pentad onset time. This is also consistent with the conclusions in our study that atmospheric aridity (high T, high VPD, and P deficit) creates a flash drought-prone environment and VPD is a significant factor influencing the speed of onset, which tends to drive a rapid decline in SM to cause flash drought events.

In addition, we extended the above analysis to the global scale to further confirm the contribution of VPD to the apparent increasing trend in flash droughts with the 1-pentad onset time. Fig. R10 shows the temporal evolutions of percentages of flash droughts with the 1-pentad onset time relative to all flash droughts for all remaining regions, except for NEU and MED. We find that all 19 regions show an increasing trend in the percentages of flash droughts

with the 1-pentad onset time, and the increasing trends for Amazon Basin (AMZ), Alaska (ALA), southern Africa (SAF), and Southeast Asia (SEA) are statistically significant ($P < 0.01$) in at least two datasets. By contrast, the increasing trends for Western Africa (WAF) and Tibet (TIB) are statistically insignificant for all three datasets (Fig. R10). Then, we compared the annual mean VPD, annual mean T, and annual total P for these regions (Fig. R11). We also find that the annual mean VPD shows an increasing trend over the regions with significant increase trends in the percentages of 1-pentad onset flash droughts, accompanied by a rise in temperature and a high probability of precipitation falling (three out of four regions show a decrease), whereas the annual mean VPD shows a decrease over WAF and TIB where the percentages of 1-pentad onset flash droughts do not increase significantly. These results further confirm the findings of our study. According to the reviewer's constructive comment, we have added the above discussions in the revised version of the manuscript.

Fig. R7 Comparison of meteorological conditions between different phases of flash droughts in light of flash droughts captured based on SM from Noah, CLSM, and GLEAM models. a–c Differences of T, VPD, and P between before (two pentads prior to flash drought) and onset phases of flash droughts in light of flash droughts captured based on SM from Noah model. d–f Differences of T, VPD, and P between onset and recovery phases of flash droughts in light of flash droughts captured based on SM from Noah model. g–i Same as a–e but for the CLSM model. m–r Same as a–e but for the GLEAM model.

Fig. R8 Comparison of percentages of flash droughts at different onset times relative to all flash droughts over NEU and MED based on GLEAM, Noah, and CLSM models. a–c Temporal evolutions of percentages of flash droughts with the onset times of 1-, 2-3, and 4-5 pentads relative to all flash droughts over NEU. **d–f** Temporal evolutions of percentages of flash droughts with the onset times of 1-, 2-3, and 4-5 pentads relative to all flash droughts over MED. The linear annual trends in the proportion of flash droughts at different onset times are estimated based on the Sen’s slope estimator, and statistical significances in trends are determined based on the MK test for the entire study period (2000–2020).

Fig. R9 Temporal evolutions of annual mean VPD, annual mean T, and annual total P over NEU and MED. a–c annual mean VPD, annual mean T, and annual total P over NEU. **d–f** Same as **a–b** but for MED. The linear annual trends in the annual mean VPD, annual mean T, and annual total P are estimated based on the Sen’s slope estimator, and statistical

significances in trends are determined based on the MK test for the entire study period (2000–2020).

Fig. R10 Comparison of percentages of 1-pentad onset flash droughts relative to all flash droughts for different regions based on GLEAM, Noah, and CLSM models. The linear annual trends in the proportion of 1-pentad onset flash droughts are estimated based on the Sen’s slope estimator, and statistical significances in trends are determined based on the MK test for the entire study period (2000–2020).

Fig. R11 Temporal evolutions of annual mean VPD, annual mean T, and annual total P over AUS, AMZ, SSA, ALA, SAF, and SEA. The linear annual trends in the annual mean VPD, annual mean T, and annual total P are estimated based on the Sen’s slope estimator, and statistical significances in trends are determined based on the MK test for the entire study period (2000–2020).

Specific comments:

Comment #1: Line 23) you should specify what has an “increasing trend globally”.

Response: We regret for the unclear statement. According to the reviewer’s comment, we have revised the statement in the revised version of the manuscript.

“And 33.64–46.18% of flash droughts developed in 5 days, with a significant ($P < 0.01$) increasing trend in the proportion of flash droughts with the 1-pentad onset time globally during the period 2000–2020.”

Comment #2: Abstract) one important finding that is omitted from the abstract is that flash droughts don't appear to be occurring more frequently in most global regions, just coming on faster.

Response: According to the reviewer's constructive suggestion, we have added this important finding in the revised version of the manuscript.

“We find that 33.64–46.18% of flash droughts developed in 5 days, with a significant ($P < 0.01$) increasing trend in the proportion of flash droughts with the 1-pentad onset time globally during the period 2000–2020. And flash droughts do not appear to be occurring more frequently in most global regions, just coming on faster. In addition, atmospheric aridity is likely to create a flash drought-prone environment, and the joint influence of soil moisture depletion and atmospheric aridity further accelerates the rapid onset of flash droughts.”

Comment #3: Line 27) I think you should delete “as a new drought type”, because it is a term that has been around for nearly 20 years now.

Response: According to the reviewer's suggestion, we have deleted “as a new drought type” in the revised version of the manuscript.

“Flash droughts have been receiving increasing attention from the scientific community in recent years.”

Comment #4: Lines 45-46) edit to "flash droughts as proposed by Mo...".

Response: According to the reviewer's suggestion, we have rewritten this sentence in the revised version of the manuscript.

“Flash droughts as proposed by Mo and Lettenmaier^{13,14} used different combinations of thresholds of soil moisture (SM), temperature (T), precipitation (P), and evapotranspiration to identify the heat wave flash drought (HWFD) and the P deficit flash drought (PDFD).”

Comment #5: Line 49) add “definition” after “Another”.

Response: We appreciate the reviewer's careful review, and have added “definition” after “Another” in the revised version of the manuscript.

“Another definition is based on the rapid intensification rate of flash droughts^{4,9,15,16}.”

Comment #6: Line 106) delete "that have a large impact on the environment". The abnormally dry class is mainly used to delineate areas that are going into or coming out of drought. If there are significant environmental impacts, the region is typically classified at a more severe level (i.e., D1, D2).

Response: We appreciate the reviewer's constructive suggestion, and have deleted "that have a large impact on the environment" in the revised version of the manuscript.

“To satisfy the drought condition, SM should decrease to a critical level (the 20th percentile threshold, which is recommended by the U.S. Drought Monitor) because SM below such a threshold indicates a start of abnormally dry conditions.”

Comment #7: Line 154) I think the term “lead time” is imprecise. You're really looking at the intensification time of the flash droughts, whereas lead time implies some sort of forecasting component.

Response: We agree with the reviewer that “lead time” is imprecise and cannot express the intensification time of flash droughts, and we think the “onset time” is more appropriate than “lead time”. Thus, we have replaced all “lead time” with “onset time” in the revised version of the manuscript.

Comment #8: Lines 176-177) I understand the label of 1-pentad flash droughts, but unfortunately it implies a flash drought that lasts 1 pentad (i.e., a duration of 1 pentad). I don't have a good solution for this issue, maybe labeling “1-pentad onset flash droughts”? This may seem pedantic, but there has been quite a lot of poor science and miscommunication around flash drought in recent years, so it's important to be as clear and precise as possible.

Response: We appreciate the reviewer’s constructive comment. For better clarification, we have replaced all “1-pentad flash droughts” with “1-pentad onset flash droughts” in the revised version of the manuscript.

Comment #9: Line 212) replace "region" with "continent".

Response: According to the reviewer’s suggestion, we have replaced “region” with “continent” in the revised version of the manuscript.

“These two places cover most of the European continent, and the annual frequencies of flash droughts of Northern Europe and Mediterranean Basin show an increasing trend (Supplementary Fig. 9).”

Reviewers' Comments:

Reviewer #2:

Remarks to the Author:

Thanks for the reply and explanation in regards to my earlier comments, edits and suggestions. I do think the paper is in a much better place.

That said, in your response to my concern about representing trends or saying a flash drought starts and can onset in 1-pentad you replied discussing that you were seeing typical drought "durations" of 5-6 pentads.

I'm fine with that, but saying it is a drought in 5 days (and the word flash "drought" is specifically used) is misleading and the abstract immediately points out that 33-46% of flash "droughts" developed in 5 days. You also base your increasing 1-pentad trend of flash "droughts" increasing statement right up front. You don't mention the "duration" and your explanation provided to me in your response.

Perhaps the wording/implication should be crafted to say rapid onset dryness at this point and drought/impacts may manifest and occur later on at 5-6 pentads (as you state in your response to me) at which point we are talking drought. After 5 days, we can not be sure how it will manifest itself going forward. There may be deterioration and there may be improvement. So using the word "drought" at 1-pentad is quite misleading without validation and impacts to back it up.

Apologies for being stuck on these semantics (drought is always hard to define), but I think it sends the wrong message about what drought is. I'm totally on board (and intimately involved) with the flash drought topic and applications so I think this paper is close, but not clear enough between rapid onset dryness and longer-term drought in the context of flash drought. I'm simply unaware of any impact I've ever seen due to drought that can happen at 1-pentad or maybe even two pentads either in natural ecosystems or agricultural systems depending on season and timing with a given phenological cycle. Unless associated with other features like heat waves or high winds, impacts take longer to manifest to a point where we would call it a drought.

As I read/interpret it, perhaps you are saying for those rapid onset dry periods/pentads that led to an eventual drought out at the 5-6 pentad time frame (along with your trend statement), you are retrospectively looking back and identifying them as droughts as they manifested themselves into a drought later on? If that is the message then this needs to be clarified in both the abstract and body of the paper. I think this would clarify things for me and the eventual readers.

Reviewer #3:

Remarks to the Author:

I very much appreciate the authors' genuine willingness to revise and supplement their analysis in response to comments by the reviewers (including myself). I am happy to recommend acceptance of the revised manuscript. Great work!

Black font = Reviewers' comments

Blue font = Response to reviewers' comments

Red font = Changes in the text of the manuscript

Response to Reviewer #2:

General comment:

Thanks for the reply and explanation in regard to my earlier comments, edits and suggestions. I do think the paper is in a much better place. That said, in your response to my concern about representing trends or saying a flash drought starts and can onset in 1-pentad you replied discussing that you were seeing typical drought "durations" of 5-6 pentads. I'm fine with that, but saying it is a drought in 5 days (and the word flash "drought" is specifically used) is misleading and the abstract immediately points out that 33-46% of flash "droughts" developed in 5 days. You also base your increasing 1-pentad trend of flash "droughts" increasing statement right up front. You don't mention the "duration" and your explanation provided to me in your response. Perhaps the wording/implication should be crafted to say rapid onset dryness at this point and drought/impacts may manifest and occur later on at 5-6 pentads (as you state in your response to me) at which point we are talking drought. After 5 days, we cannot be sure how it will manifest itself going forward. There may be deterioration and there may be improvement. So, using the word "drought" at 1-pentad is quite misleading without validation and impacts to back it up. Apologies for being stuck on these semantics (drought is always hard to define), but I think it sends the wrong message about what drought is. I'm totally on board (and intimately involved) with the flash drought topic and applications so I think this paper is close, but not clear enough between rapid onset dryness and longer-term drought in the context of flash drought. I'm simply unaware of any impact I've ever seen due to drought that can happen at 1-pentad or maybe even two pentads either in natural ecosystems or agricultural systems depending on season and timing with a given phenological cycle. Unless associated with other features like heat waves or high winds, impacts take longer to manifest to a point where we would call it a drought.

As I read/interpret it, perhaps you are saying for those rapid onset dry periods/pentads that led to an eventual drought out at the 5-6 pentad time frame (along with your trend statement), you are retrospectively looking back and identifying them as droughts as they manifested themselves into a drought later on? If that is the message, then this needs to be clarified in both the abstract and body of the paper. I think this would clarify things for me and the eventual readers.

Response: We appreciate the reviewer's careful review and regret for the unclear statement of "drought" at 1-pentad (5 days) that may lead to misunderstanding. In fact, previous studies have suggested that the definition of "flash drought" should account for both its rapid onset (i.e., flash) and the actual condition of moisture limitation (i.e., drought) (Otkin et al., 2014; Otkin et al., 2018; Liu et al., 2020; Pendergrass et al., 2020; Bolles et al., 2021). Thus, our definition takes into account both the rapid onset and the actual drought condition. The

statement of “a drought in 5 days” in the abstract means a rapid onset of drying (for 5 days) that led to an eventual drought out at the 5–6 pentads or even longer, as the reviewer mentioned in the last paragraph of his/her comments. To avoid any confusion, we have revised this sentence in the abstract and have also clarified the identification of droughts with different onset time, especially droughts with the 1-pentad onset time in the methods, body, and abstract of the paper according to the reviewer’s comment.

In the methods of the paper, we have further elaborated on the definition of flash droughts, to better clarify that the onset time refers to a rapid onset dry period before drought condition, and after such a rapid onset dry period the actual persistent condition of moisture limitation represents drought. We have emphasized that the onset time of flash droughts represents the rapid onset dry period, and after such a rapid onset of drying the drought/impacts may manifest and occur later on at 5–6 pentads or even longer. We have added the relevant statements in the revised version of the manuscript. To better clarify the difference between the rapid onset of drying and drought condition in the definition of flash droughts, we have updated Fig. 5 as shown below.

“As shown in Fig. 5, we calculated the decline rate of SM for the onset development period at each grid point. The onset point is defined as the first point where SM is over the 40th percentile and then the SM decline begins at a rate greater than the 5th percentile. The end point of the onset development is defined as the first point where SM drops below the 20th percentile. Therefore, the onset development phase of flash droughts starts from the onset point (t_o) and terminates at the end point (t_e). **The period from t_o to t_e is the timescale of the onset development phase, which is defined as the onset time of flash droughts, suggesting a rapid onset of drying. The period from t_e to t_p refers to the duration under drought condition, and during such period drought impacts may manifest and occur with a relatively long duration.** To adequately reflect drought intensity and duration that can diminish crop productivity and yield, SM should not only drop below the 20th percentile but also last for at least three pentads ($t_p - t_e \geq 3$). This criterion can also be used to exclude those events that decrease from above the 40th percentile to below the 20th percentile, and then recover up to the 40th percentile quickly (see Supplementary Fig. 19, a number of events experienced such a rapid decline in SM but recovered quickly, which should be excluded). **Since the identification of flash droughts should take into account both flash (a rapid onset of drying, $t_o - t_e$) and drought severity (SM drops below a specific threshold for a period of time, $t_e - t_p$), the whole duration of flash droughts begins from t_o to t_p .**”

Fig. 5 Schematic representation of the method used to identify a flash drought event. a Schematic representation of the whole phase of a flash drought event. SM decreases from above the 40th percentile to below the 20th percentile with an average decline rate of no less than the 5th percentile for each pentad, and SM below the 20th percentile should last for no less than 3 pentads. The blue solid line represents the 5-day mean SM percentile for a grid point. The orange and green dashed lines represent the wet (the 40th percentile at a particular time of the year during the period 2000–2020) and the dry (the 20th percentiles at a particular time of the year during the period 2000–2020) conditions of SM, respectively. The purple shaded area represents the onset development of flash droughts. **b** Schematic representation of the onset phase of a flash drought event.

In the body of the paper, when the word “onset time” appears for the first time, we have emphasized that the onset time of flash droughts represents the rapid onset dry period, and after such a rapid onset of drying the drought/impacts may manifest and occur later on at 5–6 pentads or even longer. In addition, we have replaced “1-pentad onset flash droughts” with “flash droughts with 1-pentad onset of drying” to avoid any confusion in the revised version of the manuscript.

“To conduct a comprehensive assessment of the onset time of flash droughts, we divided flash droughts into five different types according to the longest possible onset development phase of flash droughts (≤ 1 month was proposed by Otkin et al.⁴), including 1 pentad, 2 pentads, 3 pentads, 4 pentads, and 5 pentads. **Please note that the onset development phase refers to a rapid onset of drying before drought conditions occur.**”

In the abstract of the paper, we have revised the relevant statement as follows.

“The emergence of flash drought has attracted widespread attention due to its rapid onset. However, little is known about the recent evolution of flash droughts in terms of the speed of onset and the causes of such a rapid onset phase of flash droughts. Here, we present a comprehensive assessment of the onset development of flash droughts and the underlying mechanisms on a global scale. We find that 33.64–46.18% of flash droughts **with 5-day onset of drying**, and there is a significant increasing trend in the proportion of flash droughts with the 1-pentad onset time globally during the period 2000–2020. Flash droughts do not appear to be occurring more frequently in most global regions, just coming on faster. In addition, atmospheric aridity is likely to create a flash drought-prone environment, and the joint influence of soil moisture depletion and atmospheric aridity further accelerates the rapid onset of flash droughts.”

Please note that this paper aims to explore the rapid onset phase of flash droughts and underlying causes of the rapid onset which is the most important characteristic of flash droughts. To this end, we highlight the most significant findings in terms of the speed of onset and potential causes, but without a description of different durations of drought conditions in the abstract due to the 150-word limit. We understand that whether it is a drought or a flash drought, the impact is mainly reflected in its longer duration, and one, two or even three pentads may not result in severe impacts, as pointed out by the reviewer. To better address the reviewer’s main concern on drought impact, we would like to clarify again that although we mainly focus on the onset development phase of flash droughts in this paper, we also pay close attention to the duration under drought condition (SM should not only drop below the 20th percentile but also last for at least three pentads) in our definition. Thus, **no matter how long the flash drought onset dry period lasts (e.g., 1 pentad), the period of drought condition (SM below the 20th percentile), excluding the rapid onset dry period, is at least three pentads (half a month) according to our definition.** We evaluated the duration of flash droughts captured based on three datasets. We find that the mean duration of flash droughts is 16.01, 13.82, and 12.30 pentads for Noah, GLEAM, and CLSM models, respectively. About 65.82% (Noah), 64.88% (GLEAM), and 60.01% (CLSM) of flash droughts last for longer than 1 month (6 pentads). Thus, the flash droughts captured in our study may be able to guarantee that these events not only experience a rapid onset of drying but also have an impact on the environment/crops because of a relative long duration under drought condition after the rapid onset dry period. Consequently, we have made great efforts to clarify and explain the difference between the rapid onset of drying and drought condition in the body and methods along with Fig. 5 (identification of flash droughts) of the revised paper.

References:

Otkin, J. A., Anderson, M. C., Hain, C. & Svoboda, M. Examining the relationship between drought development and rapid changes in evaporative stress index. *J. Hydrometeorol.* 15 (3), 938–956 (2014).

Otkin, J. A. et al. Flash droughts: a review and assessment of the challenges imposed by rapid-onset droughts in the United States. *Bull. Am. Meteor. Soc.* **99**, 911–919 (2018).

Liu, Y. et al. Two Different Methods for Flash Drought Identification: Comparison of Their

Strengths and Limitations. *J. Hydrometeor.* **21**, 691–704 (2020).

Pendergrass, A. G. et al. Flash droughts present a new challenge for subseasonal-to-seasonal prediction. *Nat. Clim. Change* **10**, 191–199 (2020).

Bolles, K. C. et al. Tree-Ring Reconstruction of the Atmospheric Ridging Feature That Causes Flash Drought in the Central United States Since 1500. *Geophys. Res. Lett.* **48**(4), e2020GL091271 (2021).

Response to Reviewer #3:

General comment:

I very much appreciate the authors' genuine willingness to revise and supplement their analysis in response to comments by the reviewers (including myself). I am happy to recommend acceptance of the revised manuscript. Great work!

Response: We appreciate the reviewer's positive comment.

Reviewers' Comments:

Reviewer #2:

Remarks to the Author:

I would like to thank the authors for their patience and willingness to address my concerns as well as modify their paper to help clarify such a tricky subject. As such, I'm happy to recommend that their paper be published given the revisions they have incorporated.

Response to Reviewer #2's comment:

General comment:

I would like to thank the authors for their patience and willingness to address my concerns as well as modify their paper to help clarify such a tricky subject. As such, I'm happy to recommend that their paper be published given the revisions they have incorporated.

Response: We sincerely appreciate all comments and suggestions raised by the reviewer, which helped us to improve the quality of the manuscript.